# The diverse club

M.A. Bertolero[1,2], B.T.T. Yeo[3,4,5,6] & M. D'Esposito[1]

A complex system can be represented and analyzed as a network, where nodes represent the units of the network and edges represent connections between those units. For example, a brain network represents neurons as nodes and axons between neurons as edges. In many networks, some nodes have a disproportionately high number of edges as well as many edges between each other and are referred to as the "rich club". In many different networks, the nodes of this club are assumed to support global network integration. Here we show that another set of nodes, which have edges diversely distributed across the network, form a "diverse club". The diverse club exhibits, to a greater extent than the rich club, properties consistent with an integrative network function—these nodes are more highly interconnected and their edges are more critical for efficient global integration. Finally, these two clubs potentially evolved via distinct selection pressures.

---

[1] Helen Wills Neuroscience Institute and Department of Psychology, University of California Berkeley, 132 Barker Hall Berkeley, Berkeley, CA 94720, USA. [2] Department of Bioengineering, University of Pennsylvania, Hayden Hall 318C, Philadelphia, PA 19104, USA. [3] Department of Electrical and Computer Engineering, National University of Singapore, Singapore 119077, Singapore. [4] Clinical Imaging Research Centre, National University of Singapore, Singapore 117599, Singapore. [5] Singapore Institute for Neurotechnology, National University of Singapore, Singapore 117456, Singapore. [6] Memory Networks Programme, National University of Singapore, Singapore 119077, Singapore. Correspondence and requests for materials should be addressed to M.A.B. (email: mbertolero@me.com)

Many complex systems—neural, the power grid, and air traffic—can be analyzed as a network with graph theory, where units (e.g., neurons or airports) and connections (e.g., axons or flight routes) are treated as nodes and edges in a graph, respectively. These systems all exhibit a community structure—nodes cluster into communities such that nodes are more strongly connected to other members of their community than to members of other communities[1–4]. Each node within one of these communities can play a distinct role in the overall network topology. In many different systems, from brains to air traffic, calculating two nodal role metrics—strength and participation coefficient —classifies network nodes based on each node's connectivity pattern within the system[5–18].

Strength is a nodal metric of the sum of a node's edges' weights. While a node's strength captures its magnitude of connectivity, it does not capture the diversity of the node's connectivity across communities in the network. The participation coefficient is a nodal metric of the diversity of each node's connections across the network's communities[10, 12]. A node's participation coefficient is maximal if it has an equal number of edges to each community in the network. Mathematically, a node's participation coefficient is independent of the node's strength, as it only measures the diversity of a node's connections across communities. Empirically, across a wide range of networks, the participation coefficient is not correlated with strength, but nodes can be high in both strength and participation coefficient[6].

Across various networks, nodes with a high strength are connected to each other at a rate greater than would be expected in a randomly organized graph[19, 20]. This subset of highly interconnected nodes is referred to as the "rich club". The rich club is thought to be critical for global communication given that these nodes have high betweenness centrality, in that, if the shortest paths between all pairs of nodes is found, many of these shortest paths involve rich club members[5, 21]. Moreover, in human brain networks, brain regions within the rich club are more likely to exhibit pathology in many neurological and psychiatric disorders compared to other brain regions[15]. In line with this empirical finding, in silico "attacks" on networks demonstrate that, when edges between nodes in the rich club are removed, global efficiency decreases (i.e., the sum of shortest paths between all nodes increases)[5]. Given these characteristics, the rich club, which has been investigated in over 200 published reports to date, is proposed to be an integrative and stable core of brain regions that coordinates the transmission of information across the network.

However, as opposed to the high magnitude of connectivity that high strength nodes exhibit, nodes with a high participation coefficient exhibit diverse connectivity. This connectivity pattern places these nodes at the topological center of the network[22], which is putatively ideal for integration and coordination. In the human brain, these nodes are also located in many different communities and where many communities are within close physical proximity[6]. These nodes appear to control or coordinate which regions are "functionally" connected during cognition, in that activity in these nodes predicts changes in the connectivity of other nodes[23, 24], particularly the connectivity between nodes in different communities during cognitive tasks[25]. In humans, these nodes have also been implicated in a diverse range of tasks[26, 27]. Moreover, damage to these brain regions in humans causes a decrease in the modular architecture of the human brain network[28] and widespread cognitive deficits[29]. Finally, a recent analysis showed that, during human cognition, only these brain regions exhibit increased activity if more communities are engaged in a cognitive task, which suggests that they are involved in processes that are more demanding as more communities are engaged[1]. A parsimonious explanation of these empirical findings is that nodes with high participation coefficients integrate information and coordinate connectivity between communities, which allows for modular local processing.

In summary, nodes with both a high participation coefficient (i.e., diverse club) and a high strength (i.e., rich club) have been proposed to perform integrative and coordinative functions based on their high interconnectedness, high betweenness centrality, their membership in many different communities, the impact on the network's structure when they are removed, or their activity profile during cognitive tasks. Here we show that the diverse club exhibits these putatively integrative or coordinative properties to a greater extent than the rich club. We demonstrate, with data from multiple systems, that networks contain a diverse club that is more highly interconnected than the rich club. Moreover, these two clubs are largely comprised of different nodes. We report the anatomical locations of these clubs in the human brain, the connectivity patterns of these clubs, the functional responses of both clubs in the human brain during cognitive tasks, and how damage to nodes in each club impacts the network's efficiency. Finally, we present a generative evolutionary network model that generates graphs with a diverse club but not a rich club. From these analyses, we conclude that the diverse club exhibits, to a greater extent than the rich club, properties that likely support integrative or coordinative functions. They also suggest that the diverse club and rich club have distinct roles in network communication. While our focus is mostly on human brain networks, our findings generalize to smaller biological networks and man-made networks.

## Results

**Community detection and identification of the clubs**. We analyzed structural and functional networks from multiple species—the *C. elegans*' structural and functional networks, the macaque structural network, the human functional network, the United States power grid network, and the global air traffic network (see Methods section for network construction details). In the functional networks, edges are weighted by the strength of the pearson correlation between the two nodes' time series of activity. In the structural networks, edges represent axons (*C. elegans*), white matter connections (macaque), flight routes, or power lines.

We consider both structural and functional networks, as strength can be artificially inflated in functional (i.e., correlational) networks[6]. Graph theory allows for the comparison of network organization among very different systems. While the *C. elegans* networks, the macaque network, and the human brain networks are clearly different networks, they are all biological neural networks that were shaped by evolution. Thus, we also investigated man-made networks to determine if they exhibit properties similar to biological networks.

The equation for the participation coefficient (Methods section, Eq. 5) depends on the community structure; if the detected community structure of the network varies, so will the participation coefficients. Thus, for each network, we applied community detection using nine different community detection algorithms (Methods section). In the Results, we present findings from the InfoMap algorithm. For the functional networks, each algorithm was applied at 16 different densities (0.05–0.15), as well as 16 different resolution parameters (0.4–0.8) for the Louvain Resolution method and 16 different community sizes for the Walktrap N method. For the structural networks, each algorithm was applied 16 times on each structural network, as well as 16 different resolution parameters (0.4–0.8) for the Louvain Resolution method and 16 different community sizes for the Walktrap N method. The detected community structure using the Infomap algorithm for the *C. elegans* and human networks are presented in Fig. 1a, d (Supplementary Fig. 1 shows the air traffic network).

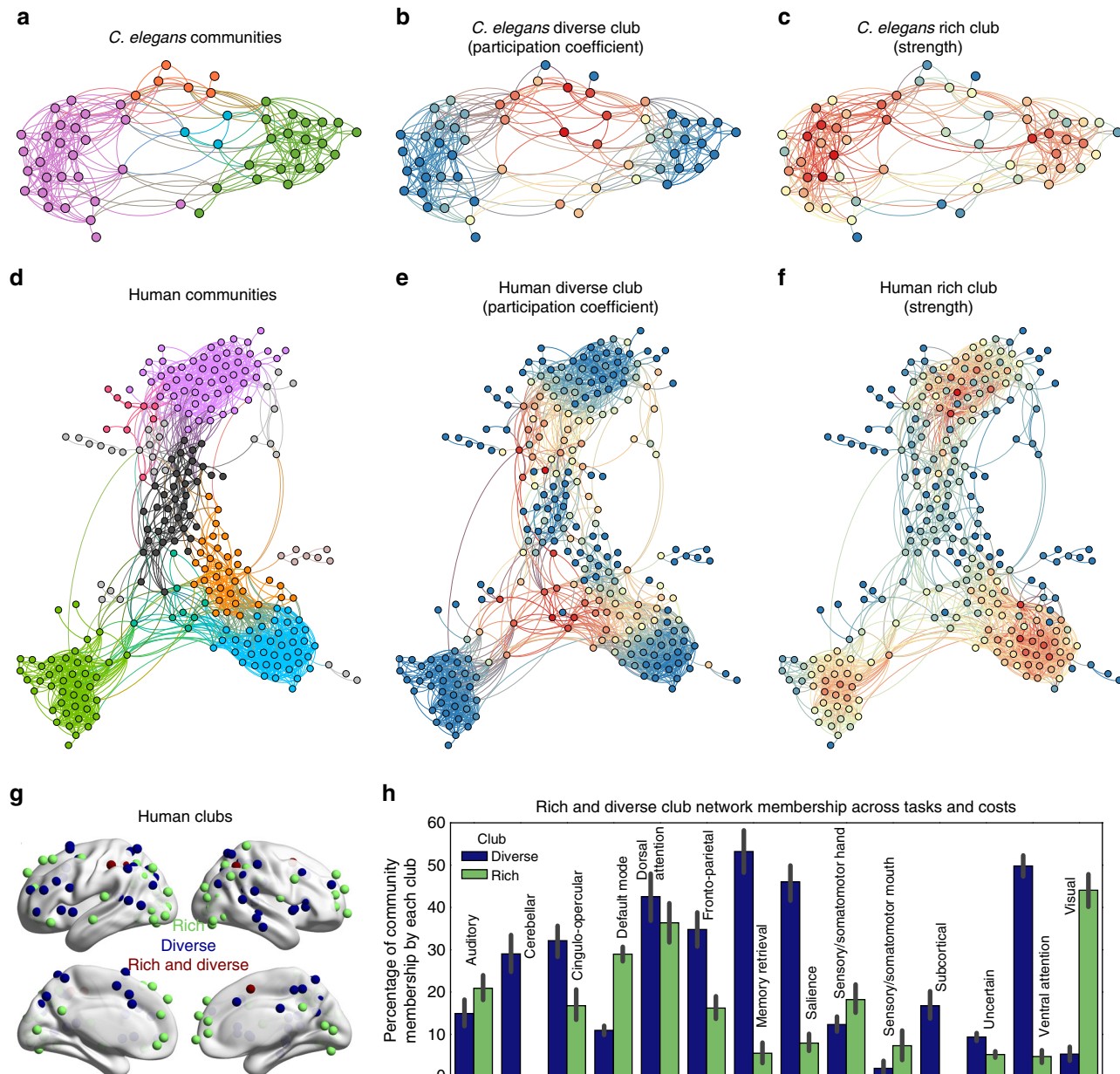

**Fig. 1** Topology of the diverse and rich clubs in the human and *C. elegans*. **a** Visualization of a single *C. elegans* functional network, labeled according to the community affiliation detected by the Infomap algorithm. **b**, **c** The *C. elegans*' diverse club and rich club. Nodes in red represent the maximum value for the given metric (participation coefficient or strength), yellow is median, and blue is the minimum. Edges are colored by the mix between the two nodes each edge connects. **d** Visualization of the human resting-state network labeled according to the community affiliation using the InfoMap algorithm. **e**, **f** The human diverse club and the rich club. In both networks, the diverse club clusters in the center of the layout, while the rich club forms clusters on the periphery. **g** The rich club and the diverse club (mean participation coefficient and strength across tasks and densities), along with nodes that are members of both clubs, are shown on the cortical surface of the human brain. **h** The mean percentage of each human cognitive system that is comprised of nodes from each club (analyzing all densities and tasks). For ease of interpretation, a canonical division of nodes into cognitive systems and names[3] are used in **h**, while all other analyses and figures use the community detection calculated here

The detected community structure was consistent across algorithms and densities or runs (Methods section; Supplementary Figs. 2–13). Next, we calculated the strength of each node in every network and visualized the distribution of those values (Supplementary Figs. 14–16). We then calculated the participation coefficient of each node in every network and visualized the distribution of those values (Methods section; Supplementary Figs. 17–31). The distribution of the participation coefficients was typically more bimodal than the strength distribution. The participation coefficients were calculated for every application of each community detection algorithm (i.e., no averaging across community detection applications, across densities, resolution parameters (Louvain), number of communities (Walktrap N), or runs was done). Overall, the participation coefficients did not dramatically vary across community detection algorithms (Supplementary Figs. 32–37). We refer to the high (80th percentile and above) strength nodes as the rich club, and the high (80th percentile and above) participation coefficient nodes club as the diverse club. For each network, the diverse club from each community detection algorithm was analyzed. Moreover, a diverse club was calculated and then analyzed for each density, resolution, run, or number of communities. A rich club was

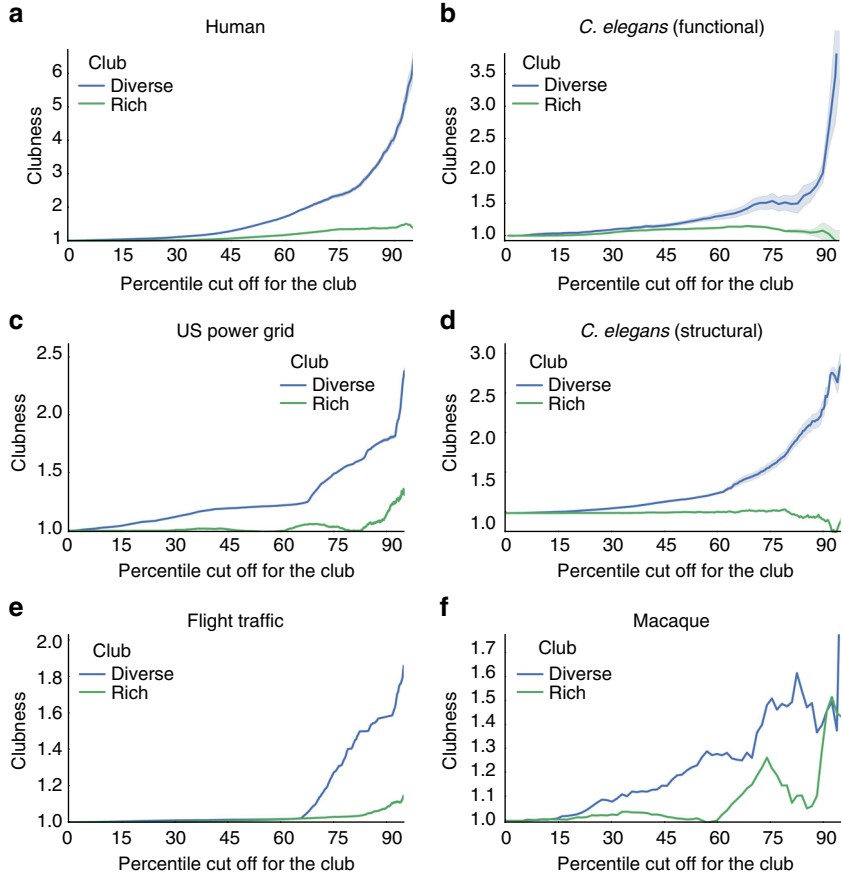

**Fig. 2** Clubness for the rich and diverse clubs in every network. **a–f** In each network, the mean across network densities (human, functional *C. elegans*) or community detection runs (macaque, structural *C. elegans*, US power grid, flight traffic) for the clubness is plotted, with 95% confidence intervals shaded. In every network, as the rank increased and only nodes with a high participation coefficient (blue) or strength (green) are included in the club, the diverse club is typically higher in clubness than that of the rich club

calculated and then analyzed for each density, as well as a rich club that utilized the unthresholded matrix ("dense rich club"). Unless otherwise stated, each individual rich club (i.e., each density) and diverse club (i.e., each density or run) is analyzed. However, we group together results (across network densities, resolution parameters, number of communities, or runs) from the same community detection algorithm. For example, the confidence intervals in Fig. 2 represent 95% confidence intervals across rich and diverse clubs of different densities or community detection runs.

**The diverse club exhibits stronger clubness than the rich club**. We refer to how interconnected a club is as "clubness" (Methods section). We measure clubness with the normalized club coefficient, which is the number of intra-club edges the club has relative to the mean of that value in a large set (here, 1000) of random graphs. These random graphs are generated based on the original graph; all nodes maintain the same number of edges and strengths, but the edges are randomly placed. For every network, we defined the rich club and its clubness across different ranks. A rank defines the cutoff for which nodes are in the rich club. For example, in a network with 100 nodes, a rank of 85 contains nodes with a strength greater than or equal to the value of the node with the 15th highest strength. In addition, for every network, we defined the diverse club—the club of high participation coefficient nodes—and its clubness at each rank. We then used multiple analyses to characterize, and make distinctions between,

the rich club (i.e., high strength nodes) and the diverse club (i.e., high participation coefficient nodes) in each network.

We sought to measure if the diverse club or the rich club is more interconnected than the other. For each network, we calculated the clubness for both clubs at every possible rank. For both clubs, for every network, as the rank increased, clubs with a clubness >1 (i.e., 1 means equal to random) were detected (Fig. 2, Supplementary Figs. 38–40). However, in every network, as the rank increased to only include those nodes with the highest strength or participation coefficient, the clubness of the diverse club was, depending on the community detection algorithm, typically equal to or higher than that of the rich club.

Results were very similar by additionally normalizing clubness by the standard deviation of the clubness values in the random graphs (Supplementary Figs. 41–43). Moreover, in weighted networks (where edges can have values different from 1) the edge weights (in the *C. elegans* functional networks, human functional networks, and air traffic network; other networks contain only binary 1 or 0 edge weights) in the random graphs can be shuffled between nodes with the same number of edges, which accounts for the contribution of both edge placement and edge weight to the normalized club coefficient[19]. Results were also very similar when additionally shuffling edge weights (Supplementary Figs. 44–46).

We also observed a positive relationship between (x) the minimum strength or participation coefficient value in the club and (y) the club's clubness value (Supplementary Figs. 47–49).

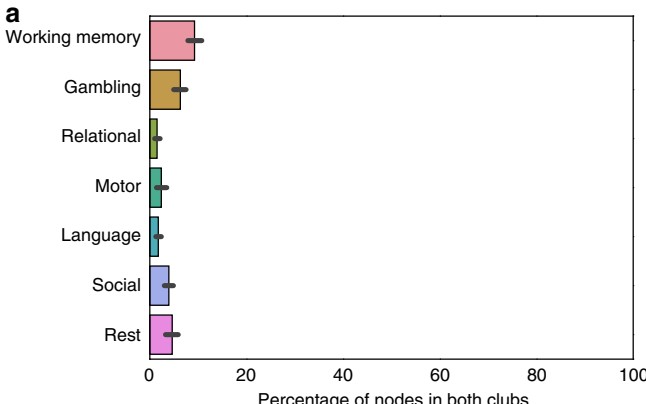

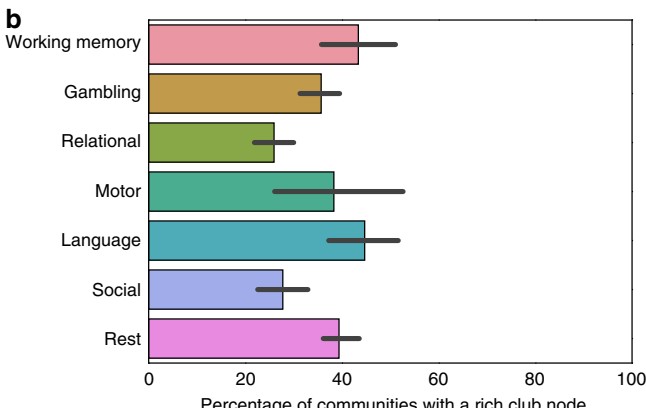

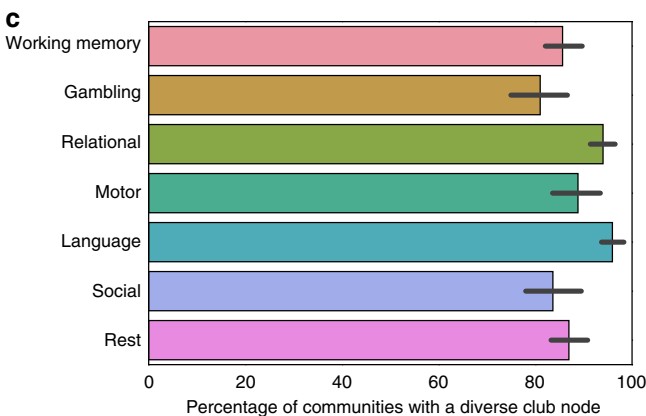

**Fig. 3** Distribution of nodes in clubs and communities. **a** The percentage of nodes in human networks that are in both the rich and diverse clubs. Zero represents that no nodes were members of both clubs, and 100% represents that the clubs are identical. **b, c** The percentage of communities in the network that contain a rich club node **b** or a diverse club node **c**. For each task, we calculate this for each density, utilizing the community detection results from that density. Thus, error bars represent the distribution of results across different densities

However, the relationship was more logarithmic for the participation coefficient value. Along with the previous finding that the participation coefficient distribution is more bimodal than the strength distribution, this suggests that membership in the diverse club is more binary than membership in the rich club. In sum, these results demonstrate that, across a range of networks, the club of high participation coefficient nodes (the diverse club) is more strongly interconnected than the club of high strength nodes (the rich club).

**The rich and diverse clubs are mostly non-overlapping.** We further analyzed clubs at the rank that corresponds to the 80th percentile, as this is the rank, across networks, where the normalized club coefficient increased dramatically (while some networks' rich club did not exhibit above chance clubness at these ranks, we still analyzed the highest strength nodes, as these nodes might exhibit other integrative properties besides high clubness). For example, in the human brain networks, which contain 264 nodes, the clubs contained 53 nodes each. To visualize the topology of the derived network communities, we used the ForceAtlas2[30] algorithm, which simulates a physical system in which nodes repel each other like charged particles and edges attract their nodes like springs, which results in nodes in the same community pulling together, and different communities pulling apart from one another. We labeled each node in the graph by their community affiliation and their membership in a rich or diverse club (Fig. 1 shows the communities in *C. elegans* and human resting-state; Supplementary Fig. 1 shows a non-biological network, air traffic).

Visual inspection of the *C. elegans* (Fig. 1a) and human functional networks (Fig. 1d) suggests that rich club nodes exist on the periphery of the graph, whereas diverse club nodes are in the center. There are few nodes that are members of both clubs. Anatomically in the human brain network, the rich club and diverse club are differentially represented in different cognitive systems (Fig. 1g, h). These analyses demonstrate that the clubs exist at different anatomical locations in the human brain as well as different topological locations in the graph.

We then quantified how similar the clubs are in each network, measuring the percentage of possible overlap. Zero represents that no nodes were members of both clubs, and 100% represents that the clubs are identical. In the human networks, across all community detection algorithms and both resting-state and 6 task states, no more than 23% of nodes were in both clubs (Fig. 3, Supplementary Fig. 50). In the functional *C. elegans* networks, across worms and algorithms, the overlap ranged from 6 to 76% (Supplementary Fig. 51). The overlap in structural networks (*C. elegans*, macaque, air traffic, and US power grid networks) ranged from 18 to 63% (Supplementary Fig. 52). These analyses demonstrate that, in general, the diverse and rich clubs are predominately comprised of different nodes.

**The rich and diverse clubs exhibit topological differences.** Given that the diverse club appears to be in the topological center of complex networks, and an integrative club of nodes should have members in many different communities, we tested how many communities has a member of each club. Across all networks and algorithms (besides the Louvain (resolution) algorithm), an equal or higher percentage of communities contained a node from the diverse club than the rich club (Fig. 3, Supplementary Figs. 50–52).

We next tested if the betweenness centrality—the number of shortest paths between pairs of nodes that pass through a node— of the diverse club is higher than that of the rich club. Across all networks we analyzed, the betweenness centrality of the diverse club was not consistently significantly higher or lower than the rich club (Supplementary Figs. 53–55). Betweenness centrality, however, does not capture if the network's shortest paths traverse edges between nodes in the rich or diverse club. Thus, we measured the edge betweenness—how many shortest paths between pairs of nodes traverse a particular edge—of the edges between members of the rich club or the diverse club. With this calculation, across algorithms, in almost all networks, the edge betweenness was significantly higher for the diverse club than the

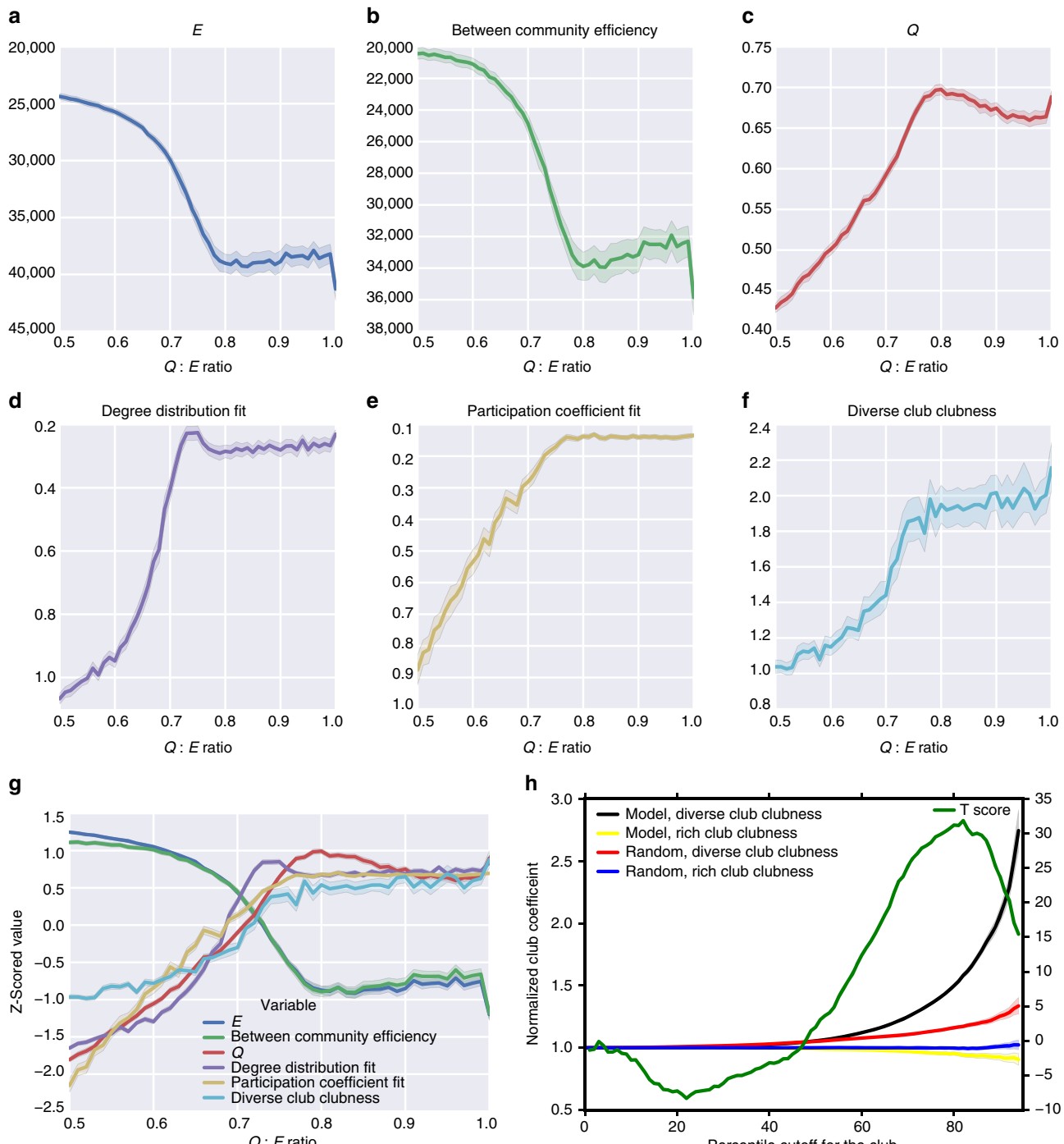

**Fig. 4** A generative modular and efficient model exhibits a diverse club. **a–f** Six features of the network were analyzed across different ratios of maximizing *Q* (modularity, see Methods section, Eqs. 1–4) and *E* (efficiency, inverse of the sum of shortest paths between all nodes, see Methods section, Eq. 6). We used the Kullback–Leibler divergence to measure the degree and participation coefficient fits; thus, lower values indicate higher similarity (see Methods section, Eq. 9). 100 models were run at each ratio in 0.01 steps. Each value's mean and 95% confidence intervals (shaded) are shown. **g** At ratios of 0.7–0.8 between weighting *Q* and weighting *E*, a balance between these six variables was achieved. **h** The average clubness across 1000 iterations for each rank for the diverse and rich clubs in the generative model at a ratio of 0.75 and the random model, as well as the *t*-test at each rank between the clubness of the diverse club in the model and the clubness of the diverse club in the random model (similar results from ratios of 0.70 and 0.80 are shown in Supplementary Fig. 66). Only the diverse club in the model has a high clubness

rich club (Supplementary Figs. 56–58). Moreover, in the air traffic network, until the clubs reached a size of 356 airports (~10% of all airports), the diverse club had more international airports in it than the rich club. Furthermore, flights between airports not in the diverse club are predominately domestic, while international flights were mainly between diverse club airports; this was not the case for the rich club and non-rich club airports (Supplementary Fig. 1). These analyses demonstrate that, relative to the rich club, the diverse club is represented in more communities and more shortest paths between nodes pass through the diverse club. These are two properties that are likely critical for global network integration and communication.

**Intra-diverse-club connections are more critical**. To further investigate the importance of the diverse and rich clubs for efficient global communication in a network, we simulated lesioning or damage to intra-club connections. For each network, in 10,000 iterations, we removed between (randomly) 50 and 90% of edges from either club (skipping edges that disconnected the graph into two sub-graphs; given that this frequently occurs in sparser networks, we used a network density of 0.20 for the functional *C. elegans* and human networks). We then calculated the increase in the sum of shortest paths between all nodes, which indicates decreased global efficiency. In every network, removing edges between diverse club nodes increased the sum of shortest paths to a greater extent than removing edges between rich club nodes (Supplementary Figs. 59–61). This demonstrates that the edges in the diverse club are more critical to efficient global communication than edges in the rich club.

**Diverse club activity increases in more complex tasks**. Previously, using the BrainMap database, we demonstrated that the diverse club (nodes with a high participation coefficient) exhibits increased activity in tasks that engaged more cognitive components or communities (see refs [1, 27] for detailed descriptions). Using the Human Connectome Database resting-state network studied here, we replicated these findings—increased activity of the diverse club was correlated with the number of communities (mean $r = 0.45$: Supplementary Fig. 62) or cognitive components (mean $r = 0.395$; Supplementary Fig. 63) a task engaged. For the rich club, nodes exhibited significantly decreased activity as more communities (mean $r = -0.37$; Supplementary Fig. 64) or cognitive components were engaged in a task (mean $r = -0.45$; Supplementary Fig. 65). Thus, the diverse club, not the rich club, exhibits increased activity when more communities are engaged in a task, which likely occurs when more integration across and coordination between communities is required. Note that, for this analysis (and this analysis only), the rich club and diverse club were defined based on the average strength or participation coefficient across densities and a previous canonical division (shown in Fig. 1d) of nodes into communities was used to estimate the number of communities engaged[3].

**The clubs potentially evolved via distinct pressures**. Evolutionary pressures have selected networks with rich clubs and diverse clubs. Thus, the final distinction between the diverse club and the rich club we sought to make is if these clubs were potentially naturally selected for different reasons. One of the first observations in neuroscience–Cajal's conservation principle—was that the brain is organized by an economic trade-off between minimizing the number of connections in the network and adaptive topological patterns[31]. One topological pattern that might be adaptive is modularity, which is how sparse the connectivity between communities is relative to the connectivity within communities. Another potentially adaptive topological pattern is efficiency, which is the inverse of the sum of shortest paths between all nodes, and thus measures how efficiently signals can be integrated across the network. For example, in brain networks, efficiency is used as a measure of the overall capacity for parallel information transfer and integrated processing[22]. Networks that are modular (i.e., exhibit high clustering) and efficient are described as "small world"[32]. Thus, we asked the question: do evolutionary pressures that select high modularity and efficiency given a limited number of connections generate a network topology that contains a rich club or a diverse club? In other words, is one of the clubs nature's solution to efficient integrative processing in a modular network?

To answer this question, we developed a generative graph model that jointly maximizes modularity, which we define as $Q$ (see Methods, Eqs. 1–4), and efficiency ($E$; the inverse of the sum of shortest paths between all nodes, see Methods, Eq. 6). The model starts with a graph of 100 nodes that are randomly connected, with 5% of all possible edges (247 binary edges). To simulate natural selection of high $Q$ and $E$, we found individual edges that, when removed, both increase $Q$ the most and decrease $E$ the least (Methods section). We remove these edges and then randomly place them back in the network, thus artificially selecting edges in the graph that jointly maximize $Q$ and $E$. We also ran the same model, except we randomly selected the edges, removed them, and then randomly placed them in the network. This allowed us to decipher if the model selects a network with a diverse club that is more highly interconnected than if random selection had occurred. Our hypothesis was that, if the diverse club is nature's solution to efficient integrative processing in a modular network, a highly interconnected diverse club, but not a rich club, will emerge when networks are selected based on maximizing modularity ($Q$) and efficient integration ($E$).

We varied the amount of importance $Q$ or $E$ played in the selection of edges. A ratio of 0.5 equally maximized both $Q$ and $E$, while 1 maximized only $Q$. We found that, at a ratio of 0.75 ($Q$)–0.25 ($E$), a balance was achieved with a high clubness of the diverse club, high $E$, high between community efficiency (the inverse of the sum of shortest paths between nodes in different communities), high $Q$, a high correspondence between the degree distribution of the model network and the human brain network (resting-state), and a high correspondence between the participation coefficient distribution of the model network and the human brain network (resting-state). To measure the correspondence between the degree and participation coefficient distributions, we calculated the Kullback–Leibler divergence between the two binned distributions for the degree or participation coefficient in the model and in the human resting-state data (see Methods section, Eq. 9). Figure 4 shows all of these metrics individually (a–f) and together (g) across different ratios of maximizing $Q$ versus $E$.

Using a ratio of 0.75, we ran 1000 iterations of the model and 1000 iterations of the random model. We then calculated the clubness of the diverse club in the model and the random model. We found that, at higher ranks, the clubness of the diverse club in the models that maximize $Q$ and $E$ was higher than the clubness of the diverse club in the random models (Fig. 4; ratios of 0.70 and 0.8 led to similar results (Supplementary Fig. 66)). This demonstrates that the diverse club's high clubness is a not a mathematical necessity of defining the club based on high participation coefficients, as randomly selected networks do not exhibit a highly interconnected diverse club. Thus, the diverse club's strong interconnectedness is a non-trivial feature of real networks. Moreover, we did not find high clubness of the rich club in the model. Thus, while the diverse club was captured by the generative model, the rich club was not captured by this model. These results suggest that the diverse club, but not the rich club, might be nature's solution to efficient integrative processing in a modular network.

## Discussion
Nodes in a network with many edges (i.e., high strength nodes) or with edges that are diversely distributed across communities (i.e., high participation coefficient nodes) are both proposed to be integrative or coordinative hubs[5–18]. Here, we provided evidence that high participation coefficient nodes, which we call the diverse club, have properties that are more characteristic of integrative hubs, as compared to high strength nodes (i.e., the rich club). The

diverse club is more interconnected than the rich club in every network we analyzed—the human brain (in 7 different tasks), the *C. elegans*, the macaque brain, the United States power grid, and global air traffic. In the human brain, diverse club nodes are up to four times as interconnected as rich club nodes. Importantly, in all networks examined, few nodes are members of both clubs.

Having established that the diverse club is relatively distinct from the rich club, we further differentiated the functions of these two clubs. The diverse club typically spans more communities, has betweenness centrality equal to the rich club, and higher edge betweenness than the rich club. This pattern of connectivity, which is spread across the entire network and exhibits the most economical route between nodes, is a critical property of nodes that integrate across network communities. Moreover, across all networks, edges between diverse club nodes are more critical to efficient global communication than the edges between rich club nodes. When diverse club edges were removed, the sum of shortest paths between nodes increased significantly more than when rich club edges were removed. Finally, in humans, these two clubs exhibit different activity patterns as cognitive tasks become more complex. Unlike rich club nodes, diverse club nodes increase activity in response to more communities being engaged by a task, which likely requires more integration across the network's communities.

We also investigated if the diverse and rich club might have distinct evolutionary origins. Many of the brain's network properties that are related to integration are heritable and impact its fitness—how likely that brain network architecture is to be naturally selected. Specifically, the brain network's cost-efficiency ratio (high efficiency given a constrained number of connections—the wiring cost) is heritable. Moreover, the diverse club's efficiency (the inverse of the sum of the shortest paths between all nodes) is heritable[33]. Efficiency is also behaviorally relevant, making it likely to factor in natural selection. For example, working memory performance is correlated with network efficiency, and individuals with schizophrenia have lower efficiency and working memory performance[34]. Also, higher intelligence quotient scores are associated with higher network efficiency and betweenness centrality of the fronto-parietal network (which we found to have the highest number of diverse club nodes)[35–37]. However, brain networks are not purely optimized for efficiency, given that they exhibit high modularity, with segregated communities performing distinct functions, at the cost of lower efficiency[38–40]. Modularity likely increases fitness in information processing systems[41–43] and confers robustness to network dynamics (i.e., information processing) when the connections between nodes are reconfigured, a process necessary for the evolution of a network[44]. Modular networks also outperform (i.e., solve tasks faster and more accurately) and evolve faster than non-modular networks[45] with lower wiring costs than non-modular networks[46]. Like efficiency, modularity is also behaviorally relevant, and thus potentially naturally selectable. For example, modularity explains intra-individual variation in working memory capacity[47] and predicts how well an individual will respond to cognitive training[48, 49].

As modularity and efficiency are both heritable and impact the fitness of an organism, we probed the possible evolutionary origins of the two clubs by asking if the rich or diverse club was selected to balance efficient global integration without sacrificing modularity. We found that, if we simulate natural selection for a balance between modularity and efficient integration, a highly interconnected diverse club, but not a rich club, emerges. Thus, the diverse club potentially evolved via selective pressures that favored both modularity and efficient integration. This provides further evidence for dissociable functions of these clubs. Additionally, the evolutionary generative model, compared to the random null model, produced significantly higher clubness in the diverse club. This demonstrates that the high clubness of the diverse club is only a feature of real world networks with a non-random architecture.

The interpretation of many previous network analyses could be dramatically altered in light of our findings, as our results provide a strong motivation for the consideration of both a rich and diverse club in network function. Contrary to previous proposals, we propose that the true integrative core of networks is the diverse club, not the rich club. Thus, we hypothesize that the rich club likely plays an alternative role in network function. One possibility that has been previously suggested is that the rich club maintains the stability of the dynamics of spontaneous activity. In the macaque structural network, rich club nodes exhibit very high in-degree—many white matter connections terminate on these nodes. Thus, autonomous dynamics of the rich club are largely constrained by the summary of strong rhythmic outputs from the entire network—rich club nodes stay closer to the summated and global network oscillations than other nodes and thus promote stability in the network dynamics at slower time-scales[50]. An analogy can be made in social networks where members that exhibit a high in-degree, like politicians, are "slaves to their own power", as they are only able to act in limited, and often slow, ways that mostly reflect the entire social network[50].

The functional connectivity, anatomical location, and cognitive functions of rich club nodes in humans fit with this proposal. The default-mode network (which we found has the most rich club nodes), is equidistant and maximally distant from primary sensory and motor networks based on both functional connectivity and anatomical geodesic distance[51]. Moreover, a meta-analysis of human brain imaging data showed that the default-mode network is involved in tasks unrelated to immediate stimulus input, such as daydreaming or mind-wandering[51]. These empirical findings suggest that the function of the rich club may predominately be to maintain stability in the entire network via slow processing, potentially using its high degree to integrate information at slower time scales, in contrast to the diverse club, which may act on shorter time scales. This potential distinction between the rich and diverse club warrants further investigation.

## Methods

**C. elegans network data**. Four *C. elegans* worms were imaged while executing behavior with calcium imaging, and each neuron's extracted time series of activity was made publically available[52]. In this analysis, each neuron was treated as a node, and the edge weights between nodes $i$ and $j$ represented the Pearson $r$ correlation (no Fisher transform was applied, as the original paper analyzed raw $r$ values and $r$ values were not averaged across worms) between the time series of nodes $i$ and $j$. The worms had 56, 77, 68, and 57 nodes. Each worms' graph was thresholded at a particular cost, retaining 5–20% of possible edges in 1% intervals. The maximum spanning tree (the set of edges (i.e., path) that connects all nodes with the maximum sum of edge weights possible) for each graph was calculated, and these edges were not removed in order to keep the graph connected. Community detection was applied at every cost separately. We also analyzed the structural network of the *C. elegans*[53], where we constructed a binary and undirected network of all 297 neurons and their 2359 axonal connections (i.e., no thresholding).

**Human functional MRI (fMRI) data**. Human fMRI data from 471 subjects (S500 release) during rest and the performance of six tasks from the Human Connectome Project[54] were used. For the task fMRI data, Analysis of Functional Neuroimages (AFNI)[55] was used to preprocess the images, matching traditional resting-state functional connectivity studies. The AFNI command *3dTproject* was used, passing the mean signal from the cerebral spinal fluid mask, the mean signal from the white matter mask, the mean whole-brain signal, and the motion parameters to the "-ort" options, which remove the signals via linear regression. The options "-automask", which generates the mask automatically was used. The "-passband 0.009 0.08" option, which removes frequencies outside 0.009 and 0.08, was used. Finally, the "-blur 6", was used, which smooths the images (inside the mask only) with a filter that has a width (FWHM) of 6 mm after the time series filtering. We analyzed the working memory (405 timepoints), relational reasoning (232 timepoints), motor (284), social cognition (274 timepoints), mixed math and language (316), and gambling tasks (253 timepoints). Given the short length (176 timepoints, and thus

low degrees of freedom during preprocessing) of the Emotion task, it was not included in our analyses. For the resting-state fMRI data (1200 timepoints), we used the images that were previously preprocessed with ICA-FIX. The AFNI command *3dBandpass* was used to further preprocess these images. We used it to remove the mean whole-brain signal and frequencies outside 0.009 and 0.08 (explicitly, "-ort whole_brain_signal.1D -band 0.009 0.08 -automask").

For each state (both LR and RL encoding directions were used), for each subject, the mean signal from 264 regions in the Power atlas[3] was computed. The Pearson $r$ between all pairs of signals was computed to form a 264 by 264 matrix, which was then Fisher z transformed. All subjects' matrices were then averaged. No negative correlations were included in our analyses. This matrix served as the edge weights for the graph for that particular state. The same thresholding and analyses across costs that was applied to the *C. elegans* functional networks was executed for human networks.

**Macaque structural network**. The structural network of the macaque cortex is publically available[56]. While the *C. elegans* is a micro-scale network, with individual neurons represented as nodes, the macaque network is a macro-level network, with 71 brain regions modeled as nodes and 438 white matter tracts modeled as edges. Edges were treated as undirected and binary; thus, no thresholding and analyses across costs is required.

**Man-made networks**. We analyzed air traffic patterns between 3281 airports and 531 airlines spanning the globe, where a node is an airport, and the edge weight between nodes is the number of airlines flying between them, resulting in 10,924 edges (data downloaded from OpenFlights.org). We also analyzed the United States power grid, where a node is either a generator, a transformer, or a substation ($n = 4941$), and an edge represents a power supply line ($n = 6594$). No thresholding was applied to either network. Data was downloaded from: http://konect.uni-koblenz.de/networks/opsahl-powergrid.

**Community detection**. Community detection algorithms are meant to group nodes into sets of nodes, where each set is a putative community. Each algorithm is essentially a definition of what a community is, and then an implementation that finds communities in the network based on that definition.

**Community detection methods that maximize Q**. Two community detection algorithms we used explicitly maximize $Q$, which is the fraction of edge weights within communities minus a constant (resolution parameter) times the expected fraction of such edges in a randomized null network[57, 58]. $Q$ is written analytically as follows. Consider a weighted and undirected graph with $n$ nodes and $m$ edges represented by an adjacency matrix $\mathbf{A}$ with elements

$$\mathbf{A}_{ij} = \text{edge weight between } i \text{ and } j. \tag{1}$$

Thus, the strength of a node is given by

$$k_i = \sum_j \mathbf{A}_{ij} \tag{2}$$

And modularity ($Q$) can be written as:

$$Q = \frac{1}{2m} \sum_{i \neq j} (\mathbf{A}_{ij} - \gamma p_{ij}) \delta(c_i, c_j). \tag{3}$$

Here, $p_{ij}$ is the probability that nodes $i$ and $j$ are connected in a random null network

$$P_{ij} = \frac{k_i k_j}{2m}, \tag{4}$$

$\gamma$ is the resolution parameter, and $c_i$ is the community to which node $i$ belongs to and $\delta(\alpha, \beta) = 1$ if $\alpha = \beta$ and $\delta(\alpha, \beta) = 0$ if $\alpha \neq \beta$. The spectral[59] method is conceptually similar to principal components analysis. This algorithm computes the leading eigenvector of the Modularity matrix and divides vertices according to the signs of the vector elements. The Louvain algorithm[60] also maximizes $Q$ with two steps that are iteratively repeated. To initialize the algorithm, each node is assigned to its own community. First, each node is placed in the community that maximizes the increase of $Q$. This is done until no increase in $Q$ can be achieved for any node. Next, each community is treated as a node, and the first step is repeated. The algorithm stops when the second iteration no longer increases $Q$.

**Label propagation**. Other algorithms, while they lead to networks with a high $Q$ value, do not explicitly maximize $Q$ and instead depend on intuitive definitions of communities and capitalize on local properties of the network. The label propagation[61] algorithm has an implementation that is similar to Louvain, but capitalizes on the fact that neighbors are often in the same community. The algorithm is initialized with each node given a unique "label", where the label is simply the community name (e.g., 0), and the labels are then propagated across the network. Each node in the network is given the label to which the maximum number of its neighbors have, which causes densely connected sets of nodes to have the same label. Labels are propagated in this fashion until every node in the network has a label to which the maximum number of its neighbors belongs and nodes with the same label are grouped together as communities.

**Edge betweenness**. The edge betweenness[62] algorithm defines communities by which sets of nodes are connected after certain edges are removed from the network. The algorithm capitalizes on the assumption that edges that have high edge betweenness (many shortest paths between nodes cross that edge) are likely to be edges between communities. The algorithm thus gradually removes the edge with the highest edge betweenness (recalculating edge betweenness after every removal). As edges are removed, the graph becomes unconnected (i.e., there exists nodes for which no path between them can be found). This breaks the network into unconnected sets of nodes (i.e., no path between nodes in different sets can be found). Each set of nodes is a community. Note that, for all networks, we used binary edges for this community detection algorithm, as weighted shortest paths finds the distance with the smallest sum of edge weights. This is appropriate for finding short travel routes, but not appropriate for the networks studied here. While one can implement this algorithm to utilize the inverse of the weights[63], we chose to use binary versions of the graphs for interpretational simplicity. It is not obvious that a strong connection is more or less important or efficient for path traversals than a weak one. The binary presence of a connection, on the other hand, is much easier to interpret in path traversals. This also applies to calculating the sum of shortest paths and thus efficiency ($E$, see Eq. 6). Thus, only calculations that require calculating shortest paths utilize binarized edges.

**Spin glass**. The spin glass[58] method defines the community structure of the network as the spin configuration that minimizes the energy of the spin glass with the spin states being the community indices.

**Random walk algorithms: Walktrap and InfoMap**. The walktrap algorithm[64] is based on random walks—random walks on a graph tend to get "trapped" into densely connected sets of nodes. The algorithm defines those sets as communities. Another random walk based algorithm is the InfoMap algorithm[65], which is based on how information randomly flows through the network; thus, a community is a set of nodes among which information flows quickly and easily.

**Resolution parameters: Walktrap N and Louvain (resolution)**. All of the algorithms have a resolution limit[66], in that smaller communities are not as likely to be detected. Moreover, for many networks, there likely exists different scales at which the network's community structure can be analyzed[67]. In other words, one can analyze the network with, e.g., 5 communities, or at a higher resolution of 20 communities. Neither of these scales is necessarily more valid. While manipulating the density of the network's connections leads to a variety in the number of communities in the network, this is not systematic, nor is it possible for the unweighted networks analyzed here. Moreover, the thresholding of these weighted networks is arbitrary. Thus, we chose two community detection algorithms that allow one to specify the scale at which the communities are detected and can utilize a dense network with no edges removed.

First, we used a version of the Louvain algorithm that maximizes stability, a measure of the community detection quality that is defined in terms of the statistical properties of a dynamical process taking place on the network, instead of $Q$. Here, the time-scale of the dynamical process acts as an intrinsic parameter that can uncover more or less communities[68].

Next, the Walktrap algorithm first renders the community detection result as a dendrogram. This dendrogram can be cut at any level. For the desired number of clusters, merges are replayed from the beginning of the random walk until the community vector has that many communities, or until there are no more recorded merges, whichever happens first. This allows one to explicitly specify the number of communities in the network.

**Participation coefficient**. Given a particular community assignment, the participation coefficient of each node can be calculated. The participation coefficient (PC) of node $i$ is defined as:

$$\text{PC}_i = 1 - \sum_{s=1}^{N_M} \left( \frac{K_{is}}{K_i} \right)^2 \tag{5}$$

where $K_i$ is the sum of $i$'s edge weights, $K_{is}$ is the sum of $i$'s edge weights to community $s$, and $N_M$ is the total number of communities. Thus, the participation coefficient is a measure of how evenly distributed a node's edges are across communities. A node's participation coefficient is maximal if it has an equal sum of edge weights to each community in the network. A node's participation coefficient is 0 if all of its edges are to a single community.

**Efficiency**. Efficiency (E) is the inverse of the sum of shortest paths between all nodes. As the sum of shortest paths increase, E decreases. Thus, E for network G is calculated as:

$$E_G = -\sum_{i<j \in G} d(i,j) \tag{6}$$

where $d$ is the shortest path distance between nodes $i$ and $j$. Note again that, for all analyses using E or d, we used binary edges while calculating the shortest paths ($d$), as weighted shortest paths finds the distance with the smallest sum of edge weights.

**Clubs and clubness**. Clubs are defined based on rank ordering the nodes based on the strength or the participation coefficient and taking the nodes with a strength or participation coefficient above a particular rank. For that club, the clubness coefficient $\theta$ is calculated as the ratio between the sum of edge weights between the club's nodes, $e$, and the number of possible connections between them. In undirected networks, this is:

$$\theta = \frac{e}{n(n-1)/2} \tag{7}$$

$\theta$ must then be normalized by comparison to the $\theta$ observed in randomized networks. While normalization can be calculated analytically, normalizing by the $\theta$ across a large number of random networks more accurately discounts structural correlations due to finite-size effects, as degree–degree correlations and higher-order effects, such as large cliques, exist in random networks[61]. Here, the random networks maintain the same degree and strength distribution, but the edges (and optionally edge weights) are randomly placed. $\theta_{rand}$ is simply the mean $\theta$ across these random networks[19]. The normalized clubness coefficient,$\theta_{norm}$, is thus:

$$\theta_{norm} = \frac{\theta}{\theta_{rand}} \tag{8}$$

$\theta_{norm}$, which we simply call clubness, is calculated at each rank. We also further normalized $\theta_{norm}$ by the standard deviation of $\theta$ in the random graphs (Supplementary Figs. 41–43). A publically available python module from a previous publication[19] was used for these calculations (github.com/jeffalstott/richclub).

**Generative evolutionary model**. The model starts with a graph of 100 nodes that are randomly connected, with 5% of all possible edges ($n = 247$). Binary edges were used. No initial community structure is imposed. For each iteration, Q and E are calculated. Edges are then chosen for removal that, when removed, lead to a maximal increase in Q and E. To achieve this, the change in Q and E following the removal of each edge is calculated. These two vectors are then rank ordered separately. The minimum of the ranks that would have been assigned to all the tied values is assigned to each value (i.e., "competition" ranking). Next, the two vectors are z-scored and then weighted according to the Q-ratio parameter. For example, if the Q-ratio is 0.75, the rank ordered vector of Q changes is multiplied by 0.75, and the rank ordered vector of shortest path changes is multiplied by 0.25. The sum of the two vectors is then calculated, giving a weighted sum of the two objectives, jointly maximizing Q and E for each edge.

These edges are removed and then randomly placed back in the graph, maintaining a constant density of edges (0.05). At each iteration, 5% of edges (13) are removed from the graph and then placed randomly back into the network. This process jointly maximizes Q and E. This procedure is repeated for 150 iterations, resulting in 1950 edges being shuffled. At this point, the generative model stops. To ensure the model is stable after 150 iterations, we plotted the mean and 95% confidence intervals of Q and E at each iteration. At around 80 iterations, both Q and E remain stable (Supplementary Fig. 67). Next, using the mean value across model runs ($n = 1000$), for each iteration, we calculated the absolute percentage difference between Q or E at that iteration with Q or E in each of the previous 40 iterations. We then take the mean absolute percentage change over those 40 iterations. We then plotted these values for each iteration (Supplementary Fig. 67). Finally, using those change values, we calculated the mean absolute change over the last 30 iterations, asserting that it was less than 1% for both Q (0.3%) and E (0.9%). For a null model, we also ran the generative process, except we randomly selected the edges, removed them, and then randomly placed them in the network.

The degree and participation coefficient fit was measured by the inverse of Kullback–Leibler divergence:

$$KLD(P \| K) = \sum_i P(i) \log \frac{P(i)}{K(i)} \tag{9}$$

where P is histogram of the model network's distribution and K is the histogram of the human brain network's distribution, both of which have been sorted into 10 bins, where each bin's value is the proportion of nodes in that bin. This was implemented in python as scipy.stats.entropy(model_histogram, human_brain_histogram).

**Data availability**. A python package was written to run all of the analyses and make all of the figures. It is available at www.github.com/mb3152/diverse_club.

This package utilizes iGraph for all graph theory analyses. This repository includes original network data files as well as the final calculated results. The package is completely object-oriented, which makes recalculating the results from the original data files, with any parameter adjustments, trivial. The only non-standard libraries (i.e., does not come installed with the Anaconda distribution of python) it depends on is iGraph and Seaborn.

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

## Acknowledgements

M.A.B and M.D. are supported by NIH Grant NS79698 and the National Science Foundation Graduate Research Fellowship Program under grant no. DGE1106400. B.T.T.Y. is supported by Singapore MOE Tier 2 (MOE2014-T2-2-016), NUS Strategic Research (DPRT/944/09/14), NUS SOM Aspiration Fund (R185000271720), Singapore NMRC (CBRG14nov007), NUS YIA, and Singapore NRF Fellowship (NRFNRFF2017-06).

## Author contributions

M.A.B. devised the concept and study; M.A.B., B.T.T.Y., and M.D. jointly designed the analyses; M.A.B. contributed new reagents/analytic tools; M.A.B. ran the experiments and analyzed the data; M.A.B., B.T.T.Y., and M.D. wrote the paper.

## Additional information

**Competing interests:** The authors declare no competing financial interests.

