## [Peer Review File · Nature Communications]

Reviewers' comments:

Reviewer #1 (Remarks to the Author):

This paper shows that nodes with high participation coefficient form a club in many complex networks, in a similar way that high degree nodes form rich clubs. The authors call this new club the "diverse club". They prove that: diverse club is present in many real networks; rich and diverse clubs are different (contain mostly different nodes); and diverse club seems to have an integrative functionality for the network, larger than the rich club. A generative model capable of producing a diverse club is also introduced, showing the diverse club appears when there is a concurrent evolutionary pressure for community structure and efficiency, while no rich club appears in this way.

The paper seems to have found an important structural feature of real complex networks, of wide interest, and the analysis is quite convincing about the novelty and relevance of this discovery. However, I would like the authors to consider the following issues before recommending it for publication.

1) My main concern is about the consistency of the definition of the diverse club. Before finding the diverse club, we need to perform a community detection, which by no means can be considered as a deterministic, unique or easily solved problem. This arises many questions:

- What happens if we change the community detection algorithm? Currently the authors use InfoMap, but there are many other approaches, from the mathematically grounded ones based on stochastic block models to those based on the optimization of modularity. Does the diverse club composition change significantly when we switch the algorithm?

- All community detection algorithms that try to find just one partition have a problem of "scale of description". It is well-known that modularity suffers from a resolution limit, but the same happens for Infomap, with its field-of-view problem. The idea is that all clustering algorithms define an implicit or explicit scale of description, which can be tuned with the help of a parameter (e.g. the resistance parameter for modularity or the length of the random walks for InfoMap). At certain scales you will find few large communities, at others a large number of small communities. An alternative solution is the use of hierarchical community detection methods which provide communities, sub-communities and so on. In summary, does the diverse club change significantly if you modify the resolution of the community detection algorithm? How do you set the proper scale of description? How could you deal with hierarchical community structures?

Since the community detection is ill-defined (in fact, algorithms do not find communities, what they do is to "define" communities and later find them according to their definition), the authors must do a good work to convince the reader the diverse club exists despite the fuzzy nature of the communities.

2) The comparison of diverse and rich club has been performed assuming an 80th percentile of the ranked nodes (see Figs. 2 and 3). However, some of the analyzed networks do not have a rich club (e.g. the *C. elegans* networks in Fig. 1), or the threshold should be set at different values for each network and type of club. It makes no sense to include in the comparison nodes that do not belong to either of the clubs.

3) The generative model tries to maximize at the same time modularity and efficiency. The description of the details of the method is very poor, impossible to reproduce. For example, the best option is to have a fixed planted partition from the beginning, but this is not mentioned, so I guess you are recalculating the communities at each time step. Another point: 1950 edges shuffled divided by 150 iterations means 13 edges shuffled at each iteration, which represent 5% of the 272 available edges. Where does the 0.25% come from? Moreover, how do you select these

13 edges? You say you try to maximize Q and minimize the sum of shortest path lengths (why not efficiency directly?), with a certain ratio between 0.5 and 1, but does it mean that you select separately e.g. 75% of the edges which maximize Q and 25% of the edges that minimize the sum? What do you do when the two sets have edges in common? Otherwise, do you define a weighted sum of the two objectives and select the edges according to this combined measure? I find this last option a better solution, but probably it is not what you are doing. Finally, 5% of the maximum number of edges that can be put in a network of 100 nodes is 247, not 272!

Other minor comments:

- 4) It is common to use as a reference for the calculation of the clubness not a set of randomized networks but their expected values, which can be found analytically.
- 5) Please include the equations to calculate the participation coefficient, Q and efficiency, and put the corresponding references. In fact, currently there is no name, description and reference for Q !
- 6) In Fig. 1 and others the rank is not a good choice for the horizontal axes, the percentile would be much more informative. In Fig. 2 and many others the axis is not a percentage (0% to 100%) but a fraction (0.0 to 1.0), please check axis labels, captions and main text.
- 7) Many references do not have the year of publication, please check them all.
- 8) The common name for the "weighted degree" is "strength".
- 9) The expression "myriad of analyses" is too colloquial and not accurate, please remove them all.

Reviewer #2 (Remarks to the Author):

Dear editor,

I am writing to report about "The diverse club: the integrative core of complex networks". The authors study the set of nodes with high participation coefficient (nodes with links well distributed across communities), the diverse club, in several real-world networks (including structural and functional brain networks). The properties of such set are compared with those of the rich club, the set of high-degree nodes. Their main findings are:

1. The two sets are substantially different
2. The diverse club is more internally connected than the rich club
3. There is more communities with nodes from the diverse club than the rich one
4. Nodes in the diverse club and the links between them are sometimes more relevant to ensure the efficiency of the network

Finally, the authors propose a generative model that selects networks with, simultaneously, high modularity and efficiency. They show that such generative model produces networks with a diverse club rather than a rich one. The main conclusion is that the diverse club supports important "integrative" functions in the brain.

The paper is sufficiently well written, and the material clearly exposed. Arguments and evidences in support of claims made are generally appropriate. I think the paper may deserve publication, with few caveats.

The diverse club has already been studied (lines 57-70), and the significance of the present findings is not clear to me. What I am supposed to learn about the brain by comparing the rich and the diverse club, and in particular from the fact that the former is more related to "integrative" functions?

If, indeed, the focus is on the integrative functions, why not directly study the "integrative" club, the set of nodes with maximal "integrativeness", whatever its definition is? One of the problem of this paper is that a quantitative definition of the integrative power/importance of a node is never

spelled out explicitly. In different sections, such mysterious property is implicitly connected with the betweenness of the links in a club, or with the decrease in efficiency upon removal of a link/node. Why then not study the set that maximize such quantities? By definition, they would do better in term of relevance to integration than the diverse club. This makes statements like that at lines 368-369 somewhat arbitrary.

A related issue has to do with the generative model. If the diverse club includes nodes that are (as much as possible) equally connected to all communities, and the generative models try to build communities that are efficiently connected, isn't it almost tautological that a diverse, rather than a rich, club is generated? The issue of relevance is something that I would like to see addressed.

Few more comments. In figure 1 one would need to show not just the ratio of inter-connectivity of the two clubs to that of a random baseline, but measure the differences in terms of standard deviations. A rich club coefficient of 1.15 could be a more important deviation than a diverse coefficient of 2 if the sd of the former is 0.05 and that of the latter is 1.

Fig 2 seem to suggest that nodes of the diverse club are more centrally located than those of the rich club. Later it is said that nodes in the diverse club are more represented in the communities. These two facts seem a little bit at odd, and few words of clarification would help.

A third observation is: how solid are these findings if a different community detection method is employed?

A fact that I find somewhat irrelevant is that the diverse club is more "clubby" than the rich one. Why is it important or relevant?

Finally, few minor things: i) I think the participation coefficient should be explicitly given in the main text. ii) the authors like to use the word "myriad" in reference to the number of analyses they have done and the number of networks they considered. Myriad comes from the greek word for 10,000 and means a huge number. Maybe "few", or "several" are more appropriate? Indeed, since only few networks have been analyzed, one may wonder how general the findings are, and how substantiated are the claims of generality e.g. at lines 18-19? Probably a reference to Colizza, V. and Flammini, A. and Serrano, M. A. and Vespignani, A. (2006). "Detecting rich-club ordering in complex networks". *Nature Physics*. 2: 2: 110–115 would be appropriate when introducing the normalized coefficient.

Reviewer #3 (Remarks to the Author):

The authors present a novel measure of network connectivity called the "diverse club" as a compliment to the recently popular rich club network statistic. While the rich club identifies a set of highly connected nodes that preferentially attach to other highly connected nodes, the diverse club instead identifies a set of nodes that connect to multiple communities within the network, and remain highly connected to each other. While I find this work relevant and interesting and think that it will have a significant impact in the field as a measure to compare and contrast with the rich club, I do have some concerns and comments that first need to be addressed.

1. My biggest critique of the manuscript in its current form is the complete lack of discussion/connection to the presence (or lack of) community structure in the studied networks.

The calculation of the participation coefficient only makes sense in networks that display a strong community structure, and the value of the participation coefficient is dependent on the specific partitioning of the community structure. However, the authors make no mention of the relationship between community structure and their metric. For those not familiar with the topic and want to apply this measure as some sort of black box calculation, this will pose problems. Before calculating the participation coefficient that defines the diverse club, it would be necessary to show that the detected community structure is robust. Division of the network into, for example, 3 versus 5 communities could have a drastic impact in how connections are distributed between communities, which could then drastically affect the rank order of the participation coefficient, therefore changing the membership of the diverse club. Note that in the case of the

rich club, the degree does not suffer this problem – the degree ranking of the nodes will never change. Even then, I still think that to discuss the rich club, one must also fully characterize the degree distribution. Here, proper characterization of the community structure is even more important, because different community detection methods will return different communities within the network.

It is therefore imperative that the authors more fully describe 1) the community detection method used (at least provide a small description of the type of algorithm in the methods section); 2) the number of detected communities returned (for all networks, not just the few examples in figure 2 and extended data fig 3 - the number of detected communities should also be stated beyond the color coding in the figure). As it is also unclear exactly how sensitive this measure is to the exact detected community structure, I think that for this initial presentation, another method of performing community detection should also be implemented (such as the popular modularity maximization) and the effect of the perturbation to the returned community structure should be characterized.

2. Thresholding across costs: what was the motivation for performing thresholding across costs for the networks? Community detection algorithms can be employed without thresholding the network. Also, if community detection was applied across different costs, was the participation coefficient calculated at each cost? This should change the rank ordering I would presume which would change the members of the diverse club? When was averaging performed in the results presented in the paper? Much more detail is needed in the description of the analysis. I could not reproduce this study with the information provided.

3. I think that it is also important to include plots of both the degree distribution and the participation coefficient distribution for the studied networks. Perhaps it might be better to rank the degrees and participation coefficients and plot them according to rank for easy comparison with the clubness plots presented. It seems as though one would also want to know if the increase in clubness corresponds with an increase either in the degree or in the participation coefficient. For example, if there is a large jump in the clubness, but the value of the participation coefficient remains nearly flat, I am not sure how to interpret this. Perhaps the authors could discuss what they expect the relationship to be between the values of the degree/participation coefficient and corresponding clubness?

More minor comments:

1. In the second sentence of the abstract, the example of a brain network depicts nodes as neurons and axons as edges. Although the authors do examine the *C. elegans* neuronal network, most of their analysis is done on human functional brain networks where nodes are brain regions and edges are statistical correlations. This could be confusing for those not involved in neuroscience research. Perhaps the authors could provide a better description of the functional brain networks in the introduction since they are referenced so much throughout the paper?

2. While it is clear that the authors are trying to write to a non-technical audience, when introducing a new metric, the relevant equations should be included within the paper. Please include equations, particularly for the participation coefficient, and the clubness metric presented in the paper.

3. Figure 2d: I am confused by this figure- 14 communities are described here, but the calculation of the participation coefficient was done with the set of 7 communities shown in 2b? Or is this referring to a different calculation? Also, this information should be presented as the fraction of the membership in the community – just knowing the number of diverse or rich club nodes is not informative since each of these communities can be of a different size.

4. Figure 3: Do these communities correspond to the communities used to calculate the

participation coefficient? I am confused as to how to interpret these results in comparison to figure 2d.

5. Please define that Q represents modularity when it is first used in line 266.

6. Generative model: The resting state human brain network is used to “optimize” the ratio of maximization of Q vs. efficiency. Would this same ratio reflect the properties of the other networks studied or is this model only relevant for the human functional brain networks?

7. Figure 4a: The x-axis needs a label. I assume that the title of each plot is the y-axis label?

8. Line 351: Can “outperform” be quantified? In what way?

9. Finally, just as a general comment, I think that it is perfectly fine to just state upfront that since the rich club coefficient has been particularly discussed in the neuroscience community in regards to human brain networks, that the paper will focus on applications to neuroscience. I understand that the authors are trying to present a general concept, but I felt that the paper kept bouncing between general statements about networks and specific examples of analysis applied only to human brain networks. Note that in line 49, “these brain regions” are suddenly referenced without first mentioning the brain as a reference network!

We thank the reviewers for their detailed and constructive critiques of our work. We have addressed all of the critiques and, as a result, we believe that the manuscript is much stronger.

The following is an overall summary of the changes we have made in our revised manuscript in response to the reviewers' comments:

1. Nine different community detection methods are implemented and analyzed.
 - a. The problem of community detection, along with each method, is described and cited.
 - b. Normalized mutual information between all community detection methods is reported for each network.
 - c. All analyses are reported with a diverse club for each community detection method.
 - d. Two of the community detection methods utilize the dense unthresholded matrix. Thus, for the weighted networks, analyses of the rich club in both the thresholded networks and the dense unthresholded networks are reported.
 - e. The impact of different community structures on the participation coefficients in the network is characterized.
 - f. The Q value and the number of communities are reported for every community detection run.
2. Distributions of participation coefficients and strengths are reported for each network.
3. Along with the clubness values across ranks, two other analyses of clubness are reported.
 - a. Clubness normalized by the standard deviation of clubness across the random networks is reported.
 - b. The correlation between (x) the minimum strength or participation coefficient value in the club and (y) clubness is now plotted, and the Pearson r values are reported.
4. The generative model is explained in greater detail.
5. The motivations of the analyses and the relevance of prior work are explained in greater detail.
6. Equations for clubness, Q, and participation coefficient are now included.
7. The word "myriad" has been removed throughout the manuscript.
8. We now use the term "strength" instead of "weighted degree".

Reviewer #1 (Remarks to the Author):

This paper shows that nodes with high participation coefficient form a club in many complex networks, in a similar way that high degree nodes form rich clubs. The authors call this new club the “diverse club”. They prove that: diverse club is present in many real networks; rich and diverse clubs are different (contain mostly different nodes); and diverse club seems to have an integrative functionality for the network, larger than the rich club. A generative model capable of producing a diverse club is also introduced, showing the diverse club appears when there is a concurrent evolutionary pressure for community structure and efficiency, while no rich club appears in this way.

The paper seems to have found an important structural feature of real complex networks, of wide interest, and the analysis is quite convincing about the novelty and relevance of this discovery. However, I would like the authors to consider the following issues before recommending it for publication.

1) My main concern is about the consistency of the definition of the diverse club. Before finding the diverse club, we need to perform a community detection, which by no means can be considered as a deterministic, unique or easily solved problem. This arises many questions:

- What happens if we change the community detection algorithm? Currently the authors use InfoMap, but there are many other approaches, from the mathematically grounded ones based on stochastic block models to those based on the optimization of modularity. Does the diverse club composition change significantly when we switch the algorithm?

This is a critical issue that we have addressed thoroughly in our revised manuscript. In addition to Infomap, we now have implemented 8 other community detection methods – (1) Louvain, (2) Louvain with a resolution parameter (Louvain resolution), (3) Walktrap, (4) Walktrap with cutting the dendrogram to achieve a particular number of communities (Walktrap N), (5) Edge Betweenness, (6) Spectral, (7) Label propagation, (8) Spin Glass. Thus, in our revised manuscript, we now report analyses with 9 different diverse clubs (one for each algorithm), and two rich clubs (thresholded matrices or the dense matrix). We also report the pairwise correlations between participation coefficients of every community detection method (including across different network densities or number of communities for Louvain Resolution and Walktrap N). Similarly, we report the pairwise normalized mutual information of every community detection method. Finally, Q values and the number of communities for each method are also reported. Overall, we did not find that changing the community detection algorithm changed our original reported results. The new results are highly consistent with the prior results.

The diversity of community detection methods—both in the algorithm itself and the utilization of a thresholded sparse or dense unthresholded matrix—allowed us to measure how the algorithm, the number of communities, and the potential hierarchical structure in the network impacted the participation coefficients in networks, which has never been reported before. Thus, we believe that these additional analyses will make an important methodological contribution to the literature, as well as strengthen the support for our scientific hypotheses.

- All community detection algorithms that try to find just one partition have a problem of “scale of description”. It is well-known that modularity suffers from a resolution limit, but the same happens for Infomap, with its field-of-view problem. The idea is that all clustering algorithms define an implicit or explicit scale of description, which can be tuned with the help of a parameter (e.g. the resistance parameter for modularity or the length of the random walks for InfoMap). At certain scales you will find few large communities, at others a large number of small communities. An alternative solution is the use of hierarchical community detection methods which provide communities, sub-communities and so on. In summary, does the diverse club change significantly if you modify the resolution of the community detection algorithm? How do you set the proper scale of description? How could you deal with hierarchical community structures?

Our Louvain (resolution) method now utilizes a resolution parameter and the full dense matrix (for the weighted networks). Moreover, The Louvain, Walktrap, and Edge Betweenness community detection methods are implemented in a manner that exploits the hierarchical nature of the network; all result in a dendrogram.

To further measure possible differences in the membership of the diverse club across levels in a hierarchical community structure, in the Walktrap N method, we cut the dendrogram at different levels to achieve different numbers of communities in the hierarchical community structure. Across levels, there was very little change to the participation coefficient and thus the diverse club membership. We quantify this for every network.

As we did not want to assume a single scale of description, we simply analyzed the diverse club across many different scales. We used the Louvain (resolution) and Walktrap N methods, which allow the user to select the resolution of the communities found or directly select the number of communities. For the 7 other community detection methods, sparser matrices typically resulted in more communities. We now report the number of communities for every community detection algorithm and run. We find that and report that participation coefficients are quite similar even when the number of communities differs, both within and across all of these community detection methods.

Since the community detection is ill-defined (in fact, algorithms do not find communities, what they do is to “define” communities and later find them according to their definition), the authors must do a good work to convince the reader the diverse club exists despite

the fuzzy nature of the communities.

This is a very important and nuanced point about community detection; we now make this point in our discussion of community detection algorithms. By exhausting many different community detection methods, we believe that we have demonstrated that the fuzzy nature of community detection is not a fatal issue for detecting a diverse club.

2) The comparison of diverse and rich club has been performed assuming an 80th percentile of the ranked nodes (see Figs. 2 and 3). However, some of the analyzed networks do not have a rich club (e.g. the *C. elegans* networks in Fig. 1), or the threshold should be set at different values for each network and type of club. It makes no sense to include in the comparison nodes that do not belong to either of the clubs.

We sought to measure the properties of the highest strength nodes and the highest participation coefficient nodes and analyze what properties these nodes exhibit, one of which is clubness. While high strength nodes did not exhibit high clubness in some networks, our goal was not to measure what properties the set of nodes with the highest clubness exhibit.

3) The generative model tries to maximize at the same time modularity and efficiency. The description of the details of the method is very poor, impossible to reproduce. For example, the best option is to have a fixed planted partition from the beginning, but this is not mentioned, so I guess you are recalculating the communities at each time step. Another point: 1950 edges shuffled divided by 150 iterations means 13 edges shuffled at each iteration, which represent 5% of the 272 available edges. Where does the 0.25% come from? Moreover, how do you select these 13 edges? You say you try to maximize Q and minimize the sum of shortest path lengths (why not efficiency directly?), with a certain ratio between 0.5 and 1, but does it mean that you select separately e.g. 75% of the edges which maximize Q and 25% of the edges that minimize the sum? What do you do when the two sets have edges in common? Otherwise, do you define a weighted sum of the two objectives and select the edges according to this combined measure? I find this last option a better solution, but probably it is not what you are doing. Finally, 5% of the maximum number of edges that can be put in a network of 100 nodes is 247, not 272!

Thank you very much for noticing these typographical errors. We have fixed these errors and greatly expanded the explanations of the model. Moreover, upon publication, we will also release a python module that allows for the reproduction of all results, including the generative model.

We do not start with or maintain a fixed partition, as we assume that the evolution of networks starts without a specified community structure, and that this community structure changes during evolution. Thus, we do calculate communities and the Q value after each iteration. We do select directly for efficiency, which is the inverse of the sum

of shortest path lengths between all pairs of nodes. We do define a weighted sum of the two objectives and select the edges according to this combined measure.

Other minor comments:

4) It is common to use as a reference for the calculation of the clubness not a set of randomized networks but their expected values, which can be found analytically.

While this can be done analytically, this method can overestimate the clubness of some networks, and, in these cases, a large set of random networks is a safer option¹. We had enough computing power to calculate the clubness of the random networks, so we utilized it. At the request of Reviewer 2, we now also show clubness values that are additionally normalized by the standard deviation of clubness across the random networks, of which no analytic solution currently exists.

5) Please include the equations to calculate the participation coefficient, Q and efficiency, and put the corresponding references. In fact, currently there is no name, description and reference for Q !

We now include detailed descriptions and the equations for the participation coefficient, Q , efficiency, and clubness.

6) In Fig. 1 and others the rank is not a good choice for the horizontal axes, the percentile would be much more informative. In Fig. 2 and many others the axis is not a percentage (0% to 100%) but a fraction (0.0 to 1.0), please check axis labels, captions and main text.

We now use the percentile or percentage on the X axis for these figures.

7) Many references do not have the year of publication, please check them all.

This has been fixed. Thank you for bringing this to our attention.

8) The common name for the “weighted degree” is “strength”.

We now use the term “strength” instead of “weighted degree”

9) The expression “myriad of analyses” is too colloquial and not accurate, please remove them all.

We have removed “myriad” from the manuscript.

Reviewer #2 (Remarks to the Author):

Dear editor,

I am writing to report about “The diverse club: the integrative core of complex networks”.

The authors study the set of nodes with high participation coefficient (nodes with links well distributed across communities), the diverse club, in several real-world networks (including structural and functional brain networks). The properties of such set are compared with those of the rich club, the set of high-degree nodes. Their main findings are:

1. The two sets are substantially different
2. The diverse club is more internally connected than the rich club
3. There is more communities with nodes from the diverse club than the rich one
4. Nodes in the diverse club and the links between them are sometimes more relevant to ensure the efficiency of the network

Finally, the authors propose a generative model that selects networks with, simultaneously, high modularity and efficiency. They show that such generative model produces networks with a diverse club rather than a rich one. The main conclusion is that the diverse club supports important “integrative” functions in the brain.

The paper is sufficiently well written, and the material clearly exposed. Arguments and evidences in support of claims made are generally appropriate. I think the paper may deserve publication, with few caveats.

The diverse club has already been studied (lines 57-70), and the significance of the present findings is not clear to me. What I am supposed to learn about the brain by comparing the rich and the diverse club, and in particular from the fact that the former is more related to “integrative” functions?

In previous studies, both high strength and high participation coefficient nodes are proposed to perform integrative functions for exclusive reasons. High strength nodes have been proposed to perform integrative functions because they have the greatest sum of edges' weights, are tightly interconnected, have high betweenness centrality, and global efficiency decreases when edges between high strength nodes are removed. High participation coefficient nodes have been proposed to perform integrative functions because they are present in and connected to many different communities and these nodes increase in activity during cognitive functions that require many communities. In this study, we directly compare these two sets of nodes to determine which is more likely to sub-serve integrative functions. The reviewer asks – why is it important to compare the network function of these two sets of nodes and what will this tell us generally about brain function?

Our goal here was to take each of these putative integrative properties mentioned above and measure which set of nodes (diverse club vs rich club) exhibit them to a

greater extent. This is critical, as it has simply been assumed that the rich club exhibits the highest clubness, the highest edge betweenness, and that global efficiency decreases the most when edges between high strength nodes are removed. No previous study has compared these metrics in the rich club versus the diverse club. Thus, the common claim (and assumption) in the neuroscience literature is that the rich club performs integrative functions because it exhibits these properties. However, as we have demonstrated in this study, this claim is false.

The upshot is that, assuming these are the properties that make nodes integrative, the nodes with diverse connectivity across the brain (high participation coefficients), not many connections (high strength), are the integrative nodes in the brain (and, it appears, in many networks). This is an important, new finding that suggests that a single and easily identifiable connectivity pattern can identify integrative nodes. This finding is also somewhat counterintuitive in that integration does not necessarily rely on how many edges a node has, but how those edges are spread across communities in the network.

Why is it important to know which nodes in the brain are integrative and which are not? First, in the healthy brain, understanding nodal roles provides significant insight into brain function. Neuroscience has carved the brain into sub-networks that play distinct roles in cognition²⁻⁵. However, understanding how information is integrated across these sub-networks has proven difficult. We believe that a better understanding of the function of the diverse club, and the clear distinction we make between the diverse and rich clubs, can lead to a more complete understanding of how the brain integrates, as well as segregates, information processing during complex cognition. Second, these findings have significant clinical implications as well. In patients with brain damage it has been shown that characterizing nodal roles predicts the extent of their deficits, the extent of their recovery of function, and how well they will respond to rehabilitation^{6,7}.

If, indeed, the focus is on the integrative functions, why not directly study the “integrative” club, the set of nodes with maximal “integrativeness”, whatever its definition is? One of the problem of this paper is that a quantitative definition of the integrative power/importance of a node is never spelled out explicitly. In different sections, such mysterious property is implicitly connected with the betweenness of the links in a club, or with the decrease in efficiency upon removal of a link/node. Why then not study the set that maximize such quantities? By definition, they would do better in term of relevance to integration than the diverse club. This makes statements like that at lines 368-369 somewhat arbitrary.

We completely agree with the reviewer that a principled definition of “integration” does not exist in the neuroscience literature despite its widespread use. We do know, however, that complex cognitive processes could not occur if no information is integrated across segregated networks. Thus, for our purposes, we analyze network properties that would likely enable information integration.

A full justification of the properties that might facilitate integration in complex networks is beyond the scope of our work—we merely used the properties that others have suggested to test our hypotheses. First, we have assumed information is likely integrated where the most information travels the most frequently. We quantify this with edge-betweenness. Second, we assume that a strongly interconnected set of nodes that also has members in many different communities is ideal for integration, as this allows those nodes to make contact with information from the distinct communities, and then utilize the high interconnectivity for integration. Third, if integration occurs between these nodes, one would expect that removing edges between these nodes would increase the sum of shortest paths between all nodes, a measure of efficient integration. Finally, integrating across communities should be more complex when there are more communities to integrate across; thus, activity at integrative nodes should increase as more communities are required for cognition. Thus, we believe the approach we have taken is reasonable for the study of network integration.

Nevertheless, the reviewer brings up an excellent point—why not just find the set of nodes that maximize those so-called integrative properties? For each of property (such as highest clubness or edge-betweenness), one could find the set of nodes that maximizes that property. Also, all of these properties could, in theory, be jointly maximized. We considered this type of analysis. However, for practical computational reasons, calculating the participation coefficient is more economical than jointly maximizing all the putative integrative properties studied here. To test all these properties, one would have to iterate through all possible combinations of club members. In a network of just 264 nodes (with 53 club members), this would mean analyzing all of the properties in 1.90516732566804^{56} different clubs. This is currently not computationally possible. Even the current analysis, ignoring the time spent on community detection and calculating the participation coefficients, took 258 hours using 40 cores and 512 gigabytes of RAM. More importantly, even if it was computationally possible, this approach does not offer obvious insight into the connectivity pattern of those nodes once they are found. For the diverse club and the rich club, we know that every node has a particular connectivity pattern—high participation coefficients or high strengths. While connectivity in the final set of nodes found under this method could be studied, there will likely be high variability—not all nodes will have high participation coefficients, for example. Thus, interpreting why this particular set of node exhibits these properties would be much more difficult.

Thus, for practical and interpretational reasons, we chose to approach the problem from the other direction. We defined the diverse club and the rich club based on a meaningful connectivity pattern that is shared across the nodes in the club—high participation coefficients or strengths—and then measured various properties that are commonly assumed to support integration or coordination in those set of nodes.

A related issue has to do with the generative model. If the diverse club includes nodes

that are (as much as possible) equally connected to all communities, and the generative models try to build communities that are efficiently connected, isn't it almost tautological that a diverse, rather than a rich, club is generated? The issue of relevance is something that I would like to see addressed.

Nodes with high participation coefficients are certainly a reasonable solution to building a modular network with efficiently connected communities. However, it is not tautological. One can imagine other solutions. For example, each community's nodes have links to a single node, with each community linking to a different node than the other communities, and these nodes are tightly interconnected. This would be the rich club, as these nodes will have many connections but only be linked to two communities. Moreover, at no place in our model do we force high participation coefficient nodes to be the solution, as we are simply maximizing modularity and minimizing the sum of shortest paths between nodes (i.e., efficiency). Moreover, it is certainly not tautological that the solution also includes many connections (i.e., high clubness) between high participation coefficient nodes.

Regarding the relevance, we want to know why the diverse club exists. The results from this model are consistent with the idea that the diverse club exists to achieve efficient integration in a modular network.

Few more comments. In figure 1 one would need to show not just the ratio of inter-connectivity of the two clubs to that of a random baseline, but measure the differences in terms of standard deviations. A rich club coefficient of 1.15 could be a more important deviation than a diverse coefficient of 2 if the sd of the former is 0.05 and that of the latter is 1.

Thank you for this suggestion. We agree. In our revised manuscript, we divide the clubness at each rank by the mean and then by the standard deviation of clubness in the random networks. We now present this calculation for every network. In general, the difference in clubness between the diverse club and the rich club actually increases upon this additional normalization, as the standard deviation of the diverse club clubness was typically lower in random networks.

Note that, because the standard deviation of the random networks is extremely low when many nodes are members of the club (low ranks), this increase the clubness at low ranks to the point where the y-axis range makes visualizing the difference in clubness between the rich and diverse club at higher ranks difficult. Thus, we show these results at the 50th percentile and above (i.e., we only show the higher ranks).

Fig 2 seem to suggest that nodes of the diverse club are more centrally located than those of the rich club. Later it is said that nodes in the diverse club are more represented in the communities. These two facts seem a little bit at odd, and few words of clarification would help.

While the diverse club has members in more communities than the rich club, it remains centrally located. While this seems counterintuitive, this is achieved because the diverse club members are spread more evenly across the communities and these nodes are also tightly interconnected. Because of this spread and tight interconnectedness, in the ForceAtlas layout, the diverse club ends up being positioned in the center of the network, with communities (each of which likely have a diverse club members in it) hovering around the diverse club.

A third observation is: how solid are these findings if a different community detection method is employed?

These findings are very robust; we have now found consistent results when using 9 different community detection methods. Please see our detailed response to reviewer 1, comment 1.

A fact that I find somewhat irrelevant is that the diverse club is more “clubby” than the rich one. Why is it important or relevant?

The rich club has been suggested by others to be an “integrative core” because of its clubness. The argument is that a set of nodes with many connections, many of which are between each other, is a wiring pattern that is optimal for integrative processing. Evincing this assumption, when edges that lead to high clubness are removed, global efficiency decreases. Our goal here was not to claim that a particular property like clubness is direct evidence for integration. Our goal was to consider all the network properties that others have suggested are evidence for integration, and test which sets of nodes—high strength or high participation coefficient nodes—exhibit these properties to the greatest extent. Finally, the set of high participation coefficient has previously been identified; one of our contributions is that these nodes are best defined as club, as the members are highly interconnected to each other.

Finally, few minor things: i) I think the participation coefficient should be explicitly given in the main text. ii) the authors like to use the word “myriad” in reference to the number of analyses they have done and the number of networks they considered. Myriad comes from the greek word for 10,000 and means a huge number. Maybe “few”, or “several” are more appropriate? Indeed, since only few networks have been analyzed, one may wonder how general the findings are, and how substantiated are the claims of generality e.g. at lines 18-19?

We now report the equation for the participation coefficient (as well as Q , clubness, and efficiency). However, given that we wrote this for a general audience, we only give an intuitive definition of the equation in the main text, and we report the equations in the supplementary methods. Also, we have now removed the word “myriad” from the revised manuscript—the reviewer’s point about the proper use of this word is well taken.

Probably a reference to

Colizza, V. and Flammini, A. and Serrano, M. A. and Vespignani, A. (2006). "Detecting rich-club ordering in complex networks". *Nature Physics*. 2. 2: 110–115
would be appropriate when introducing the normalized coefficient.

We now cite this paper.

Reviewer #3 (Remarks to the Author):

The authors present a novel measure of network connectivity called the “diverse club” as a compliment to the recently popular rich club network statistic. While the rich club identifies a set of highly connected nodes that preferentially attach to other highly connected nodes, the diverse club instead identifies a set of nodes that connect to multiple communities within the network, and remain highly connected to each other. While I find this work relevant and interesting and think that it will have a significant impact in the field as a measure to compare and contrast with the rich club, I do have some concerns and comments that first need to be addressed.

1. My biggest critique of the manuscript in its current form is the complete lack of discussion/connection to the presence (or lack of) community structure in the studied networks.

The calculation of the participation coefficient only makes sense in networks that display a strong community structure, and the value of the participation coefficient is dependent on the specific partitioning of the community structure. However, the authors make no mention of the relationship between community structure and their metric. For those not familiar with the topic and want to apply this measure as some sort of black box calculation, this will pose problems. Before calculating the participation coefficient that defines the diverse club, it would be necessary to show that the detected community structure is robust. Division of the network into, for example, 3 versus 5 communities could have a drastic impact in how connections are distributed between communities, which could then drastically affect the rank order of the participation coefficient, therefore changing the membership of the diverse club. Note that in the case of the rich club, the degree does not suffer this problem – the degree ranking of the nodes will never change. Even then, I still think that to discuss the rich club, one must also fully characterize the degree distribution. Here, proper characterization of the community structure is even more important, because different community detection methods will return different communities within the network.

It is therefore imperative that the authors more fully describe 1) the community detection method used (at least provide a small description of the type of algorithm in the methods section); 2) the number of detected communities returned (for all networks, not just the few examples in figure 2 and extended data fig 3 - the number of detected

communities should also be stated beyond the color coding in the figure). As it is also unclear exactly how sensitive this measure is to the exact detected community structure, I think that for this initial presentation, another method of performing community detection should also be implemented (such as the popular modularity maximization) and the effect of the perturbation to the returned community structure should be characterized.

We completely agree with all of the points made by the reviewer. We now utilize 9 different community detection methods. Please see our detailed response to reviewer #1, who had the same concerns (see our response to comment 1).

2. Thresholding across costs: what was the motivation for performing thresholding across costs for the networks? Community detection algorithms can be employed without thresholding the network. Also, if community detection was applied across different costs, was the participation coefficient calculated at each cost? This should change the rank ordering I would presume which would change the members of the diverse club? When was averaging performed in the results presented in the paper? Much more detail is needed in the description of the analysis. I could not reproduce this study with the information provided.

We thresholded the matrix for practical reasons—a fully dense matrix typically leads to a community structure with only 2 or 3 communities, which is likely not very accurate. Moreover, calculating the sum of shortest paths or betweenness measures in a densely connected network is meaningless, rendering our targeted attack analysis and our analyses of betweenness meaningless. However, now we utilize two algorithms, Louvain (resolution parameter) and Walktrap N, that utilize the full matrix (i.e., no thresholding is required) and allow the user to select the resolution of the communities found or directly select the number of communities. Note that we do not include targeted attacks or analyses of betweenness of the dense networks where no thresholding was done.

For all analyses apart from the BrainMap analysis, no averaging of participation coefficients was executed; we calculate the participation coefficient at each cost (i.e., density) and treat each cost as a separate network / rich club / diverse club in each analysis. However, we do present the clubness and betweenness results across costs. For example, the confidence intervals for clubness and betweenness measures are across network costs. More detail is now provided about these methods in our revised manuscript.

3. I think that it is also important to include plots of both the degree distribution and the participation coefficient distribution for the studied networks. Perhaps it might be better to rank the degrees and participation coefficients and plot them according to rank for easy comparison with the clubness plots presented. It seems as though one would also want to know if the increase in clubness corresponds with an increase either in the

degree or in the participation coefficient. For example, if there is a large jump in the clubness, but the value of the participation coefficient remains nearly flat, I am not sure how to interpret this. Perhaps the authors could discuss what they expect the relationship to be between the values of the degree/participation coefficient and corresponding clubness?

In our revised manuscript, we now report the strength and participation coefficient distributions for each network. The participation coefficient distribution is more bimodal than the degree distribution.

We also report plots, for each network, of the correlation between (x) the minimum strength or participation coefficient value in the club and (y) the club's clubness. In both the rich and diverse club, there was a correlation between these two values. However, in the diverse club, this relationship was more logarithmic than for the rich club—increasing the participation coefficient leads to smaller increases in clubness.

These two analyses suggest that there is a more explicit cutoff for which nodes should be in the diverse club than the rich club—at a certain point, increasing the participation coefficient of the club does little to the clubness, and there appears to be a set of nodes with similarly high participation coefficients. We thank the reviewer for these suggestions, which have led to additional insights regarding our findings.

More minor comments:

1. In the second sentence of the abstract, the example of a brain network depicts nodes as neurons and axons as edges. Although the authors do examine the *c.elegans* neuronal network, most of their analysis is done on human functional brain networks where nodes are brain regions and edges are statistical correlations. This could be confusing for those not involved in neuroscience research. Perhaps the authors could provide a better description of the functional brain networks in the introduction since they are referenced so much throughout the paper?

We agree that this is slightly confusing. However, we did not change the example, as we did not want to get into the weeds about functional connectivity while trying to introduce graph theory, as this made it even more confusing.

We do, however, make a clear distinction (at the very beginning of the results) between the structural and functional networks, spelling exactly what each edge represents in each network.

2. While it is clear that the authors are trying to write to a non-technical audience, when introducing a new metric, the relevant equations should be included within the paper. Please include equations, particularly for the participation coefficient, and the clubness metric presented in the paper.

We now include equations for clubness, participation coefficient, efficiency, and Q . As we are writing to a non-technical audience, these are presented in the supplementary methods.

3. Figure 2d: I am confused by this figure- 14 communities are described here, but the calculation of the participation coefficient was done with the set of 7 communities shown in 2b? Or is this referring to a different calculation? Also, this information should be presented as the fraction of the membership in the community – just knowing the number of diverse or rich club nodes is not informative since each of these communities can be of a different size.

We used the 14 canonical communities in Figure 2d (now 1d) for ease of interpretation. However, we use the InfoMap communities in Figure 2b (now 1b). We now make this clearer in the legend. For further clarity, we state that this is the only instance in which these canonical communities are used. Also, following your suggestion, in the revised manuscript we have changed Figure 2d (now 1d) to display the percentage of nodes in each community that are rich or diverse club members. This makes the figure much clearer.

4. Figure 3: Do these communities correspond to the communities used to calculate the participation coefficient? I am confused as to how to interpret these results in comparison to figure 2d.

These results analyze each specific cost and the community structure at that cost and do not utilize the canonical networks in Figure 2d (now 1d). For example, if 10 networks were detected, and the diverse club has members in 9 communities, we report a value of 90 percent. This has been clarified in the Figure 3 legend. We now clarify in Figure 2 (which is now Figure 1) that we only use that canonical community membership in that figure, and nowhere else.

5. Please define that Q represents modularity when it is first used in line 266.

Done.

6. Generative model: The resting state human brain network is used to “optimize” the ratio of maximization of Q vs. efficiency. Would this same ratio reflect the properties of the other networks studied or is this model only relevant for the human functional brain networks?

The model is only meant to capture human functional networks. We now make our focus on human networks more explicit.

7. Figure 4a: The x-axis needs a label. I assume that the title of each plot is the y-axis label?

Correct—the title of each plot is the y-axis. The x-axis label has been added.

8. Line 351: Can “outperform” be quantified? In what way?

Yes, we now specify that we mean “solve tasks faster and more accurately”.

9. Finally, just as a general comment, I think that it is perfectly fine to just state upfront that since the rich club coefficient has been particularly discussed in the neuroscience community in regards to human brain networks, that the paper will focus on applications to neuroscience. I understand that the authors are trying to present a general concept, but I felt that the paper kept bouncing between general statements about networks and specific examples of analysis applied only to human brain networks. Note that in line 49, “these brain regions” are suddenly referenced without first mentioning the brain as a reference network!

Thank you for this suggestion. We now explain that we focus on human networks in our analyses and discussion, but that our results clearly generalize to other networks. We also make it clear when we are referring to human brain networks or networks in general.

1. Colizza, V., Flammini, A., Serrano, M. A. & Vespignani, A. Detecting rich-club ordering in complex networks. *Nature Physics* **2**, 110–115 (2006).
2. Bertolero, M. A., Yeo, B. T. T., Yeo, B. T. T., D'Esposito, M. & D'Esposito, M. The modular and integrative functional architecture of the human brain. *Proc. Natl. Acad. Sci. U.S.A.* **112**, E6798–807 (2015).
3. Yeo, B. T. T. *et al.* Functional Specialization and Flexibility in Human Association Cortex. *Cereb Cortex* **25**, 3654–3672 (2015).
4. Laird, A. R. *et al.* Behavioral Interpretations of Intrinsic Connectivity Networks. http://dx.doi.org/10.1162/jocn_a_00077 **23**, 4022–4037 (2011).
5. Smith, S. M. *et al.* Correspondence of the Brain's Functional Architecture during Activation and Rest. *Proc. Natl. Acad. Sci. U.S.A.* **106**, 13040–13045 (2009).
6. Warren, D. E. *et al.* Network measures predict neuropsychological outcome after brain injury. *Proc. Natl. Acad. Sci. U.S.A.* **111**, 14247–14252 (2014).
7. Gratton, C., Nomura, E. M., Pérez, F. & D'Esposito, M. Focal Brain Lesions to Critical Locations Cause Widespread Disruption of the Modular Organization of the Brain. *Journal of Cognitive Neuroscience* **24**, 1275–1285 (2012).

REVIEWERS' COMMENTS:

Reviewer #1 (Remarks to the Author):

The authors have done a great job addressing all the issues in my review, and also those of the other reviewers, thus now I can recommend its publication.

Reviewer #2 (Remarks to the Author):

I am satisfied with the extra work that the authors done.

I remain skeptical about the "diverse" coefficient being a reliable measure of anything: as remarked by all referees it does critically depends on the how communities are defined, and the resolution scale adopted. Although the authors have done extensive work to prove that in the examples examined in the manuscript this is not an issue, it is easy to build example, e.g., in which the "diverse" coefficient of a node strongly depends on the resolution scale used.

This is not necessarily a fault that can be directly imputed to the authors: I take at face value their claim that the "diverse" coefficient is widely adopted in literature.

I have no further objections to publications

Reviewer #3 (Remarks to the Author):

The authors have done a great deal of extra analysis to more accurately characterize their methods, and now include results over a variety of community detection methods. However, given that the methods are not described in the main manuscript, it is essential that the Supplemental Materials are well organized and correctly indicate the methodology. I have a few minor comments about the main text that should be addressed and more extensive comments regarding the supplemental materials.

Comments:

Pg1:

In many different systems, from brains to air traffic, calculating two nodal role metrics—strength and participation coefficient — to the graph representing the system classifies nodes based each node's connectivity pattern.

Awkward sentence. Do you mean:

In many different systems, from brains to air traffic, calculating two nodal role metrics—strength and participation coefficient —classifies network nodes based each node's connectivity pattern within the system.

Pg2:

Evidencing the rich club's criticality, in humans, these brain regions are more likely to exhibit pathology in many neurological and psychiatric disorders compared to other brain regions.

Awkward sentence. Instead:

For example, in human brain networks, brain regions within the rich club are more likely to exhibit pathology in many neurological and psychiatric disorders compared to other brain regions.

Figure 1:

There is no "d" label in the figure and the "d" in the caption is not bold.

Pg 4:

However, we group together results (across network densities, resolution parameters, number of communities, or runs)

I believe you mean “and” not “or” since there is only one plot for each network? Same problem in the figure caption.

Pg 8:

There are few nodes that are members of both clubs. Anatomically in the human brain resting-state network, the rich club and diverse club are differentially represented in different communities (Figure 1c,d).

I think that it would be more clear to say “cognitive systems” as opposed to “communities” in order to avoid confusion with the actual communities detected for use in the calculation of participation coefficient.

Pg 11:

To answer this question, we developed a generative graph model that maximizes Q , a measure of the modular structure of the network, and efficiency.

This isn't entirely true. You are not independently maximizing both measures. Also, please provide the name of Q (modularity) and provide a reference to the equation. Oddly, efficiency is defined in the figure caption (but modularity is not) – it would make more sense to define both measures in the text as opposed to the caption.

Figure 4:

(a) “degree distribution fit” and “participation coefficient fit” are insufficient titles – from the text I gather that this is some measure of similarity between the degree distribution or participation coefficient in the model and human brain networks. However, since the measure isn't defined, I have no way of assessing if a low or high number indicates a high degree of similarity. It seems that a low number indicates high similarity?

(c) Please fix the legend so that it is all in one location.

Supplementary Materials

General comment: These materials need attention. Since methods are not clearly provided in the main manuscript, it is essential that they are clearly and completely described in the supplement.

Community detection:

a) Please provide a separate and labeled section for each method.

b) In the current community detection section, the equation for modularity (which is an alternate and not commonly used expression of modularity) while correct, doesn't match the text description of the quantity. Perhaps substitute with the common equation $Q = \frac{1}{2m} \sum (A_{ij} - \gamma P_{ij}) d(c_i, c_j)$? This expression also includes the resolution parameter, γ , which makes more sense given the later discussion of resolution.

c) If you are applying spectral partitioning to maximize modularity, this is implemented on the Modularity matrix, not on the Laplacian. Please check your methods. See Newman, M. E. J. (2006). Modularity and community structure in networks. *Proceedings of the National Academy of Sciences*, 103(23), 8577–8582.

d) Please clearly specify for each method if the calculation was done on a weighted or binary matrix.

e) The extension of the betweenness algorithm for weighted networks is given here: Newman, M. E. J. (2004). Analysis of weighted networks. *Physical Review E*, 70(5 Pt 2), 056131. It is not necessary to binarize the graph.

Participation Coefficient: Please make this it's own (labeled!) section. Please also assign a variable to this quantity and express it's value as an equation.

Efficiency:

a) Again, please assign this quantity a variable as write as an equation. NOTE: It is possible to calculate a weighted shortest path length based on a cost function, often chosen as the inverse of the weights (for example in information networks). See Muldoon, S. F. et al. Small-World Propensity and Weighted Brain Networks. Sci. Rep. 6, 22057; doi: 10.1038/srep22057 (2016) for a discussion.

Generative model: Why 150 iterations? Is the model stable at this point? What was your stability criterion?

Supplementary Figs 3,6,12:

Since the maximal value of Q is density dependent, I'm not sure what the point of this figure is.

Supplementary Figs 14,15:

The caption provides almost no information. What are the numbers in the legend?

General comment about supplementary figs: The figures need descriptive captions. I should be able to look at the figure and caption and gather all information about the graph. (Eg, Fig 33 – how is similarity assessed?)

Reviewer #1 (Remarks to the Author):

The authors have done a great job addressing all the issues in my review, and also those of the other reviewers, thus now I can recommend its publication.

Thank you very much for your time and positive contribution to this paper.

Reviewer #2 (Remarks to the Author):

I am satisfied with the extra work that the authors done.

I remain skeptical about the "diverse" coefficient being a reliable measure of anything: as remarked by all referees it does critically depends on the how communities are defined, and the resolution scale adopted. Although the authors have done extensive work to prove that in the examples examined in the manuscript this is not an issue, it is easy to build example, e.g., in which the "diverse" coefficient of a node strongly depends on the resolution scale used.

This is not necessarily a fault that can be directly imputed to the authors: I take at face value their claim that the "diverse" coefficient is widely adopted in literature.

I have no further objections to publications

Thank you very much for your time and positive contribution to this paper.

Reviewer #3 (Remarks to the Author):

The authors have done a great deal of extra analysis to more accurately characterize their methods, and now include results over a variety of community detection methods. However, given that the methods are not described in the main manuscript, it is essential that the Supplemental Materials are well organized and correctly indicate the methodology. I have a few minor comments about the main text that should be addressed and more extensive comments regarding the supplemental materials.

Thank you very much for your time and positive contribution to this paper.

Comments:

Pg1:

In many different systems, from brains to air traffic, calculating two nodal role metrics—strength and participation coefficient — to the graph representing the system classifies nodes based each node's connectivity pattern.

Awkward sentence. Do you mean:

In many different systems, from brains to air traffic, calculating two nodal role metrics—strength and participation coefficient —classifies network nodes based each node's connectivity pattern within the system.

Yes, thank you for this suggestion. This has been implemented.

Pg2:

Evidencing the rich club's criticality, in humans, these brain regions are more likely to exhibit pathology in many neurological and psychiatric disorders compared to other brain regions.

Awkward sentence. Instead:

For example, in human brain networks, brain regions within the rich club are more likely to exhibit pathology in many neurological and psychiatric disorders compared to other brain regions.

Thank you for this suggestion. This has been implemented.

Figure 1:

There is no "d" label in the figure and the "d" in the caption is not bold.

Thank you for noticing, this has been fixed.

Pg 4:

However, we group together results (across network densities, resolution parameters, number of communities, or runs)

I believe you mean "and" not "or" since there is only one plot for each network? Same problem in the figure caption.

We did mean "or", as each figure only collapses across one of the parameters. While Figure 1 is across densities (or community detection runs for the macaque, power grid, flight traffic, and structural c elegans networks), in the supplementary figures, we also alternatively collapse across the number of communities or the resolution parameter when we show results from all community detection methods.

Pg 8:

There are few nodes that are members of both clubs. Anatomically in the human brain resting-state network, the rich club and diverse club are differentially represented in different communities (Figure 1c,d).

I think that it would be more clear to say "cognitive systems" as opposed to "communities" in order to avoid confusion with the actual communities detected for use in the calculation of participation coefficient.

Thank you for this suggestion. We now use "cognitive systems" here and in the figure caption.

Pg 11:

To answer this question, we developed a generative graph model that maximizes Q , a measure of the modular structure of the network, and efficiency.

This isn't entirely true. You are not independently maximizing both measures. Also, please provide the name of Q (modularity) and provide a reference to the equation. Oddly, efficiency is defined in the figure caption (but modularity is not) – it would make more sense to define both measures in the text as opposed to the caption.

For clarity, we now detail both Q and E (efficiency) in the figure caption and main text, including equations and citations.

Thank you for noticing that we are jointly maximizing Q and E , so our statement was not precise and thus not accurate. We now state that “To answer this question, we developed a generative graph model that jointly maximizes modularity, which we define as Q (see Methods for equation), and efficiency (E ; the inverse of the sum of shortest paths between all nodes, see Methods for equation)”.

Figure 4:

- (a) “degree distribution fit” and “participation coefficient fit” are insufficient titles – from the text I gather that this is some measure of similarity between the degree distribution or participation coefficient in the model and human brain networks. However, since the measure isn't defined, I have no way of assessing if a low or high number indicates a high degree of similarity. It seems that a low number indicates high similarity?
- (c) Please fix the legend so that it is all in one location.

We used the Kullback-Leibler divergence between the two binned distributions (thus, lower values mean do mean more similarity). We describe this in the Methods section, but have added an explanation in the Results and figure legend for clarity. We now also include the equation in the Methods.

Supplementary Materials

General comment: These materials need attention. Since methods are not clearly provided in the main manuscript, it is essential that they are clearly and completely described in the supplement.

In addition to moving the Supplementary Methods to the Methods section, we have implemented all of the reviewer's requests.

Community detection:

- a) Please provide a separate and labeled section for each method.

We have now formatted the Methods section in this manner. Moreover, we have organized the Methods to detail the data, then the analyses, and then the generative model.

b) In the current community detection section, the equation for modularity (which is an alternate and not commonly used expression of modularity) while correct, doesn't match the text description of the quantity. Perhaps substitute with the common equation $Q = \frac{1}{2m} \sum (A_{ij} - \gamma P_{ij}) d(c_i, c_j)$? This expression also includes the resolution parameter, γ , which makes more sense given the later discussion of resolution.

We removed the alternative equation and inserted the equation you suggest (as well as the equations for A and P_{ij}), including the proper citation.

c) If you are applying spectral partitioning to maximize modularity, this is implemented on the Modularity matrix, not on the Laplacian. Please check your methods. See Newman, M. E. J. (2006). Modularity and community structure in networks. *Proceedings of the National Academy of Sciences*, 103(23), 8577–8582.

Thank you for noticing this. This description has been fixed.

d) Please clearly specify for each method if the calculation was done on a weighted or binary matrix.

Done. We specify that the only time binary edges are used is when the sum of shortest paths is calculated, which involved betweenness measures and the edge betweenness community algorithm. We note this whenever relevant.

e) The extension of the betweenness algorithm for weighted networks is given here: Newman, M. E. J. (2004). Analysis of weighted networks. *Physical Review E*, 70(5 Pt 2), 056131. It is not necessary to binarize the graph.

Thank you for noting this. We chose to use binary versions of the graphs for simplicity, as well as the fact that it is not obvious that a strong connection is more or less important for path traversals than a weak one. The binary presence of a connection, on the other hand, is much easier to interpret in path traversals. However, we now note that other options are possible.

Participation Coefficient: Please make this it's own (labeled!) section. Please also assign a variable to this quantity and express it's value as an equation.

Done and done.

Efficiency:

a) Again, please assign this quantity a variable as write as an equation. NOTE: It is possible to calculate a weighted shortest path length based on a cost function, often chosen as the inverse of the weights (for example in information networks). See Muldoon, S. F. et al. Small-World Propensity and Weighted Brain Networks. *Sci. Rep.* 6, 22057; doi: 10.1038/srep22057 (2016) for a discussion.

Done, and we have noted that it is possible to appropriately calculate this on weighted networks, including that citation.

Generative model: Why 150 iterations? Is the model stable at this point? What was your stability criterion?

We plotted the mean and 95 percent confidence intervals of Q and E (efficiency; inverse of the sum of shortest paths) at each iteration. At around 80 iterations, both Q and E remain stable. Next, using the mean value across model runs ($n=1000$), for each iteration, we calculated the absolute percentage difference between Q or E at that iteration with Q or E in each of the previous 40 iterations. We then take the mean absolute percentage change over those 40 iterations. We then plotted these values for each iteration.

Finally, using those change values, we calculated the mean absolute change over the last 30 iterations, asserting that it was less than 1 percent for both Q (0.3%) and E (0.9%). We now describe this in the Methods section and include this figure.

Supplementary Figs 3,6,12:

Since the maximal value of Q is density dependent, I'm not sure what the point of this figure is.

While Q is certainly density dependent, this is not true for all community algorithms, and the relationship between density and Q differs across algorithms. Moreover, readers might want to know what the variance in Q is across community detection methods, as well as for each network.

Supplementary Figs 14,15:

The caption provides almost no information. What are the numbers in the legend?

General comment about supplementary figs: The figures need descriptive captions. I should be able to look at the figure and caption and gather all information about the graph. (Eg, Fig 33 – how is similarity assessed?)

We have added much more detailed captions for all of the Supplementary Figures.